# MORE SPACE IS ALL YOU NEED: REVISITING MOLECULAR REPRESENTATION LEARNING

## ABSTRACT

Molecular representation learning (MRL) has become pivotal in leveraging limited supervised data for applications such as drug discovery and material design. While early MRL methods relied on 1D sequences and 2D graphs, recent advancements have incorporated 3D conformational information, focusing predominantly on atomic interactions within 3D space. However, we argue that the space beyond atoms is also crucial for MRL, which is overlooked by prior models. To address this, we propose a novel transformer-based framework, dubbed Space-Former, which incorporates additional 3D space beyond atoms to enhance molecular representation ability. SpaceFormer introduces three key components: (1) Precision-Preserved Gridding, which discretizes continuous 3D space into grid cells while preserving precision; (2) Grid Sampling, which employs an importance sampling strategy to improve efficiency; and (3) Linear-Complexity 3D Positional Encoding, which extends Rotary Positional Encoding to 3D space to capture pairwise directions and utilizes random Fourier features to efficiently encode pairwise distances. Extensive experiments show that SpaceFormer significantly outperforms previous 3D MRL models across various tasks, validating the benefit of leveraging the additional 3D space beyond atoms in MRL models.

## 1 INTRODUCTION

Molecular representation learning (MRL), or molecular pretraining, has been a key area of research for its crucial role in utilizing limited supervised data, particularly in real-world applications such as drug design and material discovery (Gilmer et al., 2017; Rong et al., 2020). The evolution of this field has progressed from 1D sequences (Xu et al., 2017; Wang et al., 2019; Heller et al., 2015) and 2D graphs (Hu et al., 2019; Rong et al., 2020; Li et al., 2021; Wang et al., 2022b) to 3D conformations (Stärk et al., 2022; Zhou et al., 2023), incorporating increasingly rich physical information and achieving superior performance. In all these prior 3D MRL models, atoms play a central role. More specifically, these models take the types and 3D positions of *atoms* (or *atom tuples*) as inputs and focus on modeling atomic interactions within 3D space, using graph neural networks or transformers (Zhou et al., 2023; Feng et al., 2023; Wang et al., 2023; Cui et al., 2024; Yang et al., 2024).

While this atom-based MRL approach appears straightforward, we argue that it has an inherent limitation: *it ignores the spaces beyond atoms*. While it might seem intuitive to assume that these empty spaces contain no valuable information and are therefore less relevant, this assumption may overlook significant physical facts. In the theory of microscopic physics, the space beyond atoms is not truly empty; it is occupied by electrons, various electromagnetic fields, and quantum phenomena (Atkins & Friedman, 2011; Zee, 2010; Weinberg, 1995). Moreover, in many computational simulation methods used in physics, it is essential to consider the entire 3D space, not just the positions of atoms. For instance, electronic density distributions and potential fields are all functions of the entire 3D space (Atkins & Friedman, 2011; Parr et al., 1979; Szabo & Ostlund, 2012).

This fact inspires us to ask the following question:

*Will leveraging the 3D space beyond atoms improve molecular representation learning?*

In this paper, we provide an affirmative answer to the above question by introducing a novel transformer-based MRL framework called SpaceFormer, illustrated in Fig. 1. Unlike previous 3D MRL approaches that focus solely on atomic positions, SpaceFormer incorporates the space beyond atoms. To achieve this, SpaceFormer features three key components for efficient and effective 3D space processing:

Figure 1: Overview of the SpaceFormer framework. This figure illustrates the model using a 2D plane for simplicity, while SpaceFormer actually operates in 3D space. Unlike previous 3D MRL models that focus solely on atomic positions, SpaceFormer integrates the space beyond atoms. It begins by discretizing the 3D cuboid around the molecules into grid cells. To enhance efficiency, a grid sampling strategy is applied to reduce the number of input cells. Despite the discretization, precise atomic positions are retained by incorporating in-cell positions as additional input features. Moreover, SpaceFormer utilizes 3D Directional Positional Encoding with RoPE (3D Directional PE with RoPE) and 3D Distance Positional Encoding with Random Fourier Features (3D Distance PE with RFF) to effectively encode pairwise positional relationships in 3D space.

1. *Precision-Preserved Gridding*: To efficiently process the continuous 3D space, we discretize it into a grid composed of two types of cells: atom cells and non-atom cells. To mitigate the precision loss associated with discretization, in-cell positions are additionally utilized for atom cells.

2. *Grid Sampling*. Even with grid discretization, the entire grid remains too large for efficient processing. To address this, we propose an importance sampling strategy for non-atom cells, which enhances efficiency without compromising accuracy.

3. *Linear-Complexity 3D Positional Encoding*: We extend Rotary Positional Encoding (RoPE) (Su et al., 2024) to 3D continuous space to efficiently capture pairwise directional information. Additionally, we use random Fourier features to approximate Gaussian kernels (Rahimi & Recht, 2007) on pairwise distances, enabling efficient encoding of radial distance information.

With these three key components, SpaceFormer efficiently and effectively processes discretized grid cells. Extensive experiments demonstrate its superior performance compared to previous 3D MRL models across a variety of downstream tasks. Ablation studies further validate that each component plays a critical role in enhancing SpaceFormer's performance and efficiency. Additionally, we extend the Uni-Mol (Zhou et al., 2023) baseline by incorporating empty points, revealing that empty space can also benefit atom-based models. However, compared to SpaceFormer, the performance gains are significantly smaller, and Uni-Mol struggles with the efficient handling of large number of empty points. Together, these findings underscore the effectiveness of leveraging 3D space beyond atomic positions and highlight the superior performance of the proposed SpaceFormer.

## 2    RELATED WORK

**Molecular Representation Learning**    Molecular representation learning has explored various modalities, resulting in diverse methods utilizing different molecular information. Some approaches use 1D sequences, such as SMILES-BERT (Wang et al., 2019) and Xu et al. (2017). Others focus on 2D topologies; for example, MolCLR (Wang et al., 2022b), MolGNet (Li et al., 2021), Hu et al. (2019), GROVER (Rong et al., 2020). Some works further improve 2D MRL models addi-

tionally with 3D information, such as GEM (Fang et al., 2022), 3D-Infomax (Stärk et al., 2022), MoleBLEND (Yu et al., 2024), GraphMVP (Liu et al., 2021) and Transformer-M (Luo et al., 2022).

Recently, starting with Noisy Nodes (Zaidi et al., 2022) and Uni-Mol (Zhou et al., 2023), pure 3D MRL models have demonstrated superior performance across various tasks. Building on their success, recent works (Feng et al., 2023; Wang et al., 2023; Cui et al., 2024; Yang et al., 2024) have further explored the potential of 3D MRL models. While most of these efforts focus on designing new pre-training tasks based on 3D atomic positions, SpaceFormer takes a different approach by leveraging the additional empty space beyond atoms.

Apart from 3D MRL, several other domains also focus on 3D conformations, such as deep potential models (Schütt et al., 2017; Thomas et al., 2018; Gasteiger et al., 2020; 2021; Liu et al., 2022; Wang et al., 2022a; Jiao et al., 2023), protein folding (Jumper et al., 2021; Abramson et al., 2024), and 3D conformation generation (Shi et al., 2021; Zhu et al., 2022b; Xu et al., 2022; 2021). However, these works are less directly related to this paper.

**Enhancing Model Performance with Additional Tokens**   Though counterintuitive, the use of seemingly meaningless additional tokens has been shown to improve model performance in both language and vision tasks. For instance, Darcet et al. (2023) introduced register tokens into the input sequence of vision transformers, helping to mitigate artifacts and enhance performance across multiple tasks. Similarly, Pfau et al. (2024) demonstrated that using dot tokens ("...") as chain-of-thought prompts can boost large language model performance.

The concept of leveraging empty space in SpaceFormer is related to these methods but is grounded in physical principles. In particular, unlike the repeated, seemingly meaningless tokens in (Darcet et al., 2023) and (Pfau et al., 2024), SpaceFormer incorporates empty cells (non-atom cells) with distinct 3D positions, reflecting the true physical distribution of 3D space.

**Virtual Points As Intermediate Representation**   In the domain of point cloud, virtual points have been proposed as intermediate representations. For example, Wu et al. (2023); Yin et al. (2021) convert 2D camera images into 3D virtual points, which are then fused with 3D LiDAR points to create a unified input representation. Similarly, Zhu et al. (2022a) introduce sparse virtual points to align and fuse features from 2D camera images and 3D LiDAR data, effectively addressing the resolution disparity between the sensors. Numerous studies (Song et al., 2023a;b; Mahmoud et al., 2023) have further advanced this direction, utilizing virtual points as a bridge to align and fuse data from heterogeneous sensors. In contrast, molecular representation learning (MRL) tasks primarily focus on predicting molecular properties, where intermediate representations for merging heterogeneous data sources are not inherently required. Our study in Sec.4.4 demonstrates that simply adding virtual points to existing atom-based MRL models provides limited performance improvement. The contribution of SpaceFormer lies in identifying this gap and proposing a framework that effectively leverages empty space information, addressing an overlooked aspect of MRL.

## 3   SPACEFORMER

To evaluate whether incorporating the 3D space beyond atoms enhances molecular representation learning (MRL), we present SpaceFormer, a novel transformer-based MRL framework that expands beyond atomic positions. The primary challenge in this approach lies in achieving efficient implementation, since the 3D space contains an infinite number of points. To address the challenge, a common solution is grid discretization, which divides the space into discrete cells, allowing the model to process only this finite number of cells. However, this solution suffers from several drawbacks. First, even with coarse discretization, the number of cells grows cubically, which may hinder the efficiency of model training. Second, discretization can lead to a loss of precision, particularly when encoding precise atomic positions, which may negatively impact model performance.

To address these challenges, SpaceFormer incorporates 3 key components to enhance both efficiency and performance, as described below.

### 3.1   PRECISION-PRESERVED GRIDDING

In this subsection, we describe the details of grid discretization used in SpaceFormer, focusing on three key aspects.

**The Effective Cuboid for Gridding.**  We aim to ensure the cuboid fully encompasses the entire molecule while minimizing its volume as much as possible. To achieve this, we apply Principal Component Analysis (PCA) to the atomic positions to compute the three orthogonal axes according to the eigenvectors, forming a right-handed, normalized coordinate system. The atomic coordinates are then transformed into this new system, and the cuboid is defined by the circumscribed rectangular cuboid of the atoms.

**The Edge Length of Grid Cells.**  To simplify processing, we ensure that each grid cell contains at most one atom, by setting a sufficiently small cell edge length. Specifically, the cell edge length $c_l$ must satisfy $c_l < \frac{\hat{d}}{\sqrt{3}}$, where $\hat{d}$ represents the minimum Gaussian distance between any pair of atoms. Given that this paper primarily focuses on small organic molecules, $\hat{d}$ is approximately 0.96Å corresponding to the O-H bond length.

**Preserving Atomic Precise Positions after Gridding.**  Since each grid cell contains at most one atom, the cells can be categorized as either atom cells or non-atom cells. For a non-atom cell, the *cell center* represents its 3D position, while for an atom cell, the *precise positions of the atoms* are used to define the 3D position. In both cases, let $c_i \in \mathbb{R}^3$ denote the 3D position of the $i$-th cell, where $c_i$ represents the cell center for non-atom cells, and the precise atomic position for atom cells.

This position $c_i$ is used in two ways. First, it is used in global 3D positional encoding, which is detailed in Sec. 3.3. Second, it is used to compute in-cell positional features, which along with the cell type will serve as input to the SpaceFormer. Formally, the input feature for the $i$-th cell is defined as $a_i = \{t_i, e_i^0, e_i^1, e_i^2\}$, where $t_i, e_i^0, e_i^1, e_i^2 \in \mathbb{N}$ represent the cell type and the in-cell positions along the three axes, respectively. The cell type $t_i$ corresponds to the atom type for atom cells, and a special type $t_{\text{null}}$ for non-atom cells. The in-cell position $e_i$ is calculated and discretized from position $c_i$, by $e_i = \left\lfloor \frac{c_i \mod c_l}{c_m} \right\rfloor$, where $c_l$ is the cell edge length and $c_m$ is the hyper-parameter for discretization, which is set to a very small value. As a result, each $e_i$ is an integer value ranging from 0 to $\frac{c_l}{c_m}$. Finally, we convert the discrete input features $a_i$ into continuous feature representations by summing the corresponding embedding layers, denoted as $x_i = \sum_{t=0}^{3} \text{Embed}_t(a_i^t)$, where $\text{Embed}_t(\cdot)$ represents the embedding function that maps discrete inputs to continuous representations, and $x_i$ is the resulting input embedding for the $i$-th cell.

To summarize, by leveraging the above approaches, SpaceFormer achieves efficient grid discretization while preserving atomic precision.

## 3.2  Grid Sampling

Despite the efficient grid discretization described above, the number of grid cells remains too large for effective processing. For instance, in widely used organic molecule datasets like ZINC (Sterling & Irwin, 2015), the average number of cells is approximately 6,000, which makes the $\mathcal{O}(n^2)$ complexity of vanilla transformer models both computationally expensive and memory intensive. To address this challenge, we incorporate FlashAttention (Dao et al., 2022), which avoids the $\mathcal{O}(n^2)$ peak memory cost of vanilla attention, allowing for more efficient handling of larger number of cells. Additionally, we propose a sampling strategy to drastically reduce the number of non-atom cells.

Specifically, in microscopic physics, regions close to atoms exhibit higher electron density, with the density varying significantly within these regions. Consequently, computational simulations often apply fine-graining to regions near atoms to capture these dynamic variations more accurately, while coarse-graining is commonly used in regions farther from atoms to reduce computational cost. Inspired by this approach, we propose a sampling strategy based on the distance to atom cells.

Formally, for the $i$-th non-atom cell, its sampling probability is calculated as follows:

$$d_i = \min_j \left( \{ \|c_i - c_j\|_2 \mid j \in \mathcal{S}_{\text{atom}} \} \right), \ i \in \mathcal{S}_{\text{non\_atom}},$$

$$p_i = \frac{\exp(-d_i/\tau)}{\sum_{k \in \mathcal{S}_{\text{non\_atom}}} \exp(-d_k/\tau)}, \tag{1}$$

where $\tau > 0$ is the temperature for sampling, $\mathcal{S}_{\text{atom}}$ is the set of atom cells, and $\mathcal{S}_{\text{non\_atom}}$ is the set of non-atom cells. Based on the sampling probability $p_i$, we sample $m \times 100\%$ of the non-atom cells, where $m \in [0, 1]$ is a pre-defined hyper-parameter. We also perform extensive ablation studies on this sampling strategy in Sec. 4.

### 3.3 LINEAR-COMPLEXITY 3D POSITIONAL ENCODING

Positional encoding is critical in both transformer-based models and 3D MRL models. However, existing methods from these models cannot be directly applied to SpaceFormer. First, the default positional encoding in transformers is typically discrete, such as sequence order in language models, while the cell positions $c_i$ in SpaceFormer are continuous. Second, in 3D MRL models, SE(3)-invariant positional encoding (Zhou et al., 2023), often based on pairwise Gaussian distances, can be effective but are computationally inefficient because it has a memory cost of $\mathcal{O}(n^2)$ and is impractical for handling large number of grid cells.

To address these challenges, we propose an efficient positional encoding method tailored to continuous 3D coordinates by focusing on encoding pairwise positional information of grid cells. Given two 3D points, $A$ and $B$, with coordinates $c_A$ and $c_B$, respectively, their pairwise positional information is represented as $\vec{AB} = c_B - c_A$. Based on this, we propose two linear-complexity 3D positional encodings: the first directly encodes $\vec{AB}$, capturing directional information through raw positional deltas and retaining dependency on the coordinate system; the second encodes the pairwise distance $\|\vec{AB}\|_2$, which is invariant to the coordinate system.

**3D Directional Positional Encoding with RoPE** In transformer models, several types of positional encodings are commonly used (Vaswani, 2017; Dufter et al., 2022). Recently, Rotary Positional Encoding (RoPE) has become the default due to its linear-complexity in encoding relative positions. In SpaceFormer, we extend RoPE to 3D continuous space to encode directional information ($\vec{AB}$), capturing pairwise directional relationships across all three axes among cell positions.

The key idea behind RoPE is to apply a set of 2D rotation matrices to the Query and Key in the attention module, with angles dependent on positions. After performing the Query-Key dot product, the relative position between them is encoded. Formally, for an attention head with hidden dimension $d_h$, the $i$-th Query vector $q_i \in \mathbb{R}^{d_h}$ is split into $d_h/2$ tensors of length 2, with $q_{i,l} \in \mathbb{R}^2$ representing the $l$-th tensor. Similarly, $k_{j,l} \in \mathbb{R}^2$ represents the $l$-th tensor of the $j$-th Key. Then, during the dot product in the attention module, we compute:

$$q_{i,l}R_l(i)(k_{j,l}R_l(j))^T = q_{i,l}R_l(i)R_l(j)^T k_{j,l}^T = q_{i,l}R_l(i-j)k_{j,l}^T, \tag{2}$$

where $R_l(i) \in \mathbb{R}^{2\times 2}$ is the $l$-th 2D rotation matrix, with the angle depending on position $i$. Due to the group property of rotation matrices, we have $R_l(i)R_l(j)^T = R_l(i-j)$, thus encoding the relative position $i - j$.

To effectively extend RoPE to 3D continuous space, we adapt its rotation matrices to handle continuous positions. Specifically, the rotation matrix $R_l(\cdot)$ is designed to accept continuous inputs. Moreover, since there are multiple rotation matrices, we partition them across the three axes in 3D space. We divide the $d_h/2$ matrices into 3 sets, each with $c_r = \lfloor d_h/6 \rfloor$ matrices, corresponding to the 3 axes. The resulting rotation matrices are:

$$\{R_0(c_i^0), \ldots, R_{c_r-1}(c_i^0), R_{c_r}(c_i^1), \ldots, R_{c_r\times 2-1}(c_i^1), R_{c_r\times 2}(c_i^2), \ldots, R_{c_r\times 3-1}(c_i^2)\}. \tag{3}$$

If $d_h$ is not divisible by 6, the remaining $d_h/2 - c_r \times 3$ matrices will be identity matrices. In summary, this extended RoPE retains linear complexity and encodes relative continuous positions independently along each of the three axes, effectively capturing pairwise directional information in 3D space.

**3D Distance Positional Encoding with RFF** While the above RoPE-based encoding captures directional information, it inherently depends on the coordinate system. Although PCA is used to establish a coordinate system during gridding, it cannot always guarantee a unique solution, particularly in the presence of symmetry in molecular data. To address this, we additionally encode the pairwise distance ($\|\vec{AB}\|_2$), which is invariant to the coordinate system, offering a more stable representation of pairwise positional information.

However, directly encoding pairwise distances results in a high memory cost of $\mathcal{O}(n^2)$. To overcome this, following Rahimi & Recht (2007), we propose using random Fourier features (RFF) to approximate the Gaussian kernel on pairwise distances with linear complexity:

$$\exp\left(-\frac{\|c_i - c_j\|^2}{2\sigma^2}\right) \approx z(c_i)z(c_j)^T,$$

$$z(c_i) = \sqrt{\frac{2}{d_h}}\cos(c_i\frac{\omega}{\sigma} + b), \quad \omega \in \mathbb{R}^{3\times d_h} \sim \mathcal{N}(0, I), b \in \mathbb{R}^{d_h} \sim \mathcal{U}([0, 2\pi]^{d_h}), \tag{4}$$

where $\sigma$ controls the shape of the Gaussian, $\boldsymbol{\omega}$ is sampled from standard normal distribution, and $\boldsymbol{b}$ is sampled from a uniform distribution over $[0, 2\pi)^{d_h}$.

The random Fourier features are then combined with the Query and Key after applying the RoPE:

$$\boldsymbol{q}_i = f(\boldsymbol{q}_i, z(\boldsymbol{c}_i)) \quad \boldsymbol{k}_j = f(\boldsymbol{k}_j, z(\boldsymbol{c}_j)), \tag{5}$$

where $f$ represents the combination function, which can be either addition or concatenation to incorporate $z(\boldsymbol{c}_i)z(\boldsymbol{c}_j)^T$ through the dot-product in the attention module. Addition is simple and efficient, though it introduces extra noise terms in the attention, such as $\boldsymbol{q}_i z(\boldsymbol{c}_j)^T$. Concatenation avoids these noise terms but is less efficient as it doubles the dimensionality of the dot-product from $d_h$ to $2d_h$. In our early experiments, we observed no significant performance difference between the two methods, so we opted for the more efficient addition as the combination function.

### 3.4 OVERALL FRAMEWORK

Combining the above components, we outline the overall framework of SpaceFormer, as shown in Fig. 1. In summary, SpaceFormer is highly efficient: it utilizes grid discretization with importance sampling strategy, to convert infinite 3D continuous space into a manageable number of grid cells, and addresses the $\mathcal{O}(n^2)$ bottleneck in Transformers using FlashAttention and linear-complexity 3D positional encoding. In terms of effectiveness, SpaceFormer retains in-cell positions as input features, selectively samples important non-atom cells, and accurately captures pairwise 3D positional information through the proposed positional encoding.

## 4 EXPERIMENTS

To comprehensively evaluate SpaceFormer's performance, we first conduct unsupervised pretraining on large-scale unlabeled data, following previous works. The pre-trained model is then fine-tuned on various tasks with limited labeled data. Extensive ablation studies are also performed to assess the contribution of each component. Additionally, we present an in-depth comparison with atom-based models that also incorporate empty space to provide deeper insights into why SpaceFormer works.

### 4.1 SETTINGS

**Pre-training Settings** To further reduce training costs, we employ the Masked Auto-Encoder (MAE) pretraining strategy (He et al., 2022), which reduces the number of cells used during pre-training. Specifically, MAE is an encoder-decoder architecture where the encoder processes only unmasked inputs (e.g., 70% of the cells). The decoder then attempts to predict the types and in-cell positions of the masked cells based on the encoder's outputs. This approach is highly efficient because (1) the encoder processes only a subset of cells, and (2) although the decoder processes all cells, it is significantly smaller than the encoder. Furthermore, only the encoder is used during downstream task fine-tuning.

For a fair comparison, we use the same pretraining dataset as the previous work Uni-Mol (Zhou et al., 2023), which includes a total of 19 million molecules. Details of the pre-training settings are provided in Table 7 in the Appendix. For grid sampling, we set the sampling ratio $m$ to 0.1 and the sampling temperature $\tau$ to 1.0 by default unless otherwise specified. This configuration results in a model with approximately 58.7 million parameters (55.1M in the encoder) and requires about 32 hours of training using 8 NVIDIA RTX 4090 GPUs.

**Baseline Models** Our primary baseline is Uni-Mol (Zhou et al., 2023), a recent 3D MRL model that achieved state-of-the-art performance on most molecular property prediction tasks. Additionally, SpaceFormer uses the same pretraining dataset as Uni-Mol, enabling an apple-to-apple comparison. We also include Mol-AE (Yang et al., 2024), which extends Uni-Mol with MAE pretraining strategy. Furthermore, for a more comprehensive comparison, we further include two 2D graph-based MRL models: GROVER (Rong et al., 2020) and GEM (Fang et al., 2022).

**Downstream Tasks** Most prior works use MoleculeNet (Wu et al., 2018) for downstream task evaluation. However, recent studies (Walters, 2023) have identified several limitations within the MoleculeNet dataset, including the presence of invalid structures, inconsistent chemical representations, and data curation errors. Additionally, (Sun et al., 2022) has shown that MoleculeNet fails to adequately distinguish the performance of different molecular pretraining models. To address these issues, we developed a new benchmark framework to comprehensively evaluate MRL models.

Table 1: Performance on molecular computational property prediction tasks. The best results are highlighted in **bold**, and the second-best results are underlined.

| Model | mu $\downarrow$ ($D$) | alpha $\downarrow$ (Bohr$^3$) | R$^2$ $\downarrow$ (Bohr$^2$) | ZPVE $\downarrow$ (Hartree) | C$_v$ $\downarrow$ (cal/(mol*K)) | HOMO $\downarrow$ (Hartree) | LUMO $\downarrow$ (Hartree) | GAP $\downarrow$ (Hartree) |
|---|---|---|---|---|---|---|---|---|
| \multicolumn{9}{c}{In-Distribution Split} |
| GROVER | 0.6505 ± 1.7e-2 | 0.7330 ± 4.1e-2 | 42.0297 ± 8.2e0 | 0.0006 ± 5e-5 | 0.2290 ± 2e-2 | 0.0052 ± 5.3e-4 | 0.0055 ± 4.7e-4 | 0.0079 ± 1.5e-3 |
| GEM | 0.5480 ± 3.6e-3 | 0.3881 ± 2.8e-3 | 25.9474 ± 1.8e-1 | 0.0003 ± 8e-5 | 0.1514 ± 7.6e-4 | 0.0039 ± 8e-5 | 0.0041 ± 2e-5 | 0.0057 ± 4e-5 |
| Uni-Mol | 0.1552 ± 2.9e-3 | 0.1675 ± 1.4e-2 | 2.4775 ± 1.7e-1 | 0.0003 ± 5e-5 | 0.0742 ± 2.7e-3 | 0.0019 ± 2e-5 | 0.0018 ± 2e-5 | 0.0029 ± 4e-5 |
| Mol-AE | 0.1583 ± 4e-3 | 0.1697 ± 1.2e-2 | 2.8530 ± 5.4e-1 | 0.0010 ± 8e-5 | 0.0843 ± 1.2e-2 | 0.0020 ± 8e-5 | 0.0030 ± 8e-5 | 0.0040 ± 8e-5 |
| SpaceFormer | **0.0552** ± 3.6e-4 | **0.1445** ± 2.4e-3 | **1.7169** ± 3.3e-2 | **0.0001** ± 1.4e-5 | **0.0585** ± 7.4e-4 | **0.0016** ± 1.1e-5 | **0.0015** ± 1.3e-5 | **0.0026** ± 2e-5 |
| \multicolumn{9}{c}{Out-of-Distribution Split} |
| GROVER | 0.5062 ± 2.5e-3 | 0.6456 ± 1.2e-1 | 46.2615 ± 4.9e0 | 0.0008 ± 1.4e-4 | 0.2527 ± 1.4e-2 | 0.0069 ± 5.9e-4 | 0.0050 ± 4.2e-4 | 0.0069 ± 5.9e-4 |
| GEM | 0.4433 ± 9.6e-3 | 0.3577 ± 5.1e-3 | 30.8420 ± 1.8e-1 | **0.0003** ± 8e-5 | 0.1540 ± 4.2e-3 | 0.0041 ± 5e-5 | 0.0042 ± 1e-5 | 0.0061 ± 1.1e-4 |
| Uni-Mol | 0.1430 ± 1.7e-3 | 0.1761 ± 4.1e-3 | 3.8530 ± 4.4e-1 | 0.0004 ± 5e-5 | 0.0914 ± 1.9e-3 | 0.0020 ± 7e-5 | 0.0024 ± 9e-5 | 0.0034 ± 1e-5 |
| Mol-AE | 0.1457 ± 1.3e-3 | 0.1947 ± 3.5e-2 | 4.6540 ± 6.1e-1 | 0.0020 ± 8e-5 | 0.0830 ± 2.9e-3 | 0.0023 ± 4.7e-4 | 0.0033 ± 4.7e-4 | 0.0047 ± 4.7e-4 |
| SpaceFormer | **0.0493** ± 1.3e-3 | **0.1425** ± 3.1e-3 | **2.8363** ± 2.8e-2 | 0.0003 ± 1.3e-5 | **0.0675** ± 1.4e-3 | **0.0017** ± 1.3e-5 | **0.0019** ± 3.3e-5 | **0.0031** ± 3.1e-5 |

Our benchmark comprises two categories of tasks: molecular computational properties and molecular experimental properties. For computational properties, we sampled a 40K subset from the QM9 dataset (Ramakrishnan et al., 2014) and selected 8 representative properties. This sampling approach allows us to assess model performance on limited labeled data. For experimental properties, we selected the BBBP and BACE datasets from MoleculeNet, ensuring that all duplicate and structurally invalid molecules were excluded. Additionally, we employed the HLM, MDR1-MDCK ER (MME), and Solubility (Solu) datasets from the Biogen ADME dataset (Fang et al., 2023). A detailed description of these tasks is provided in Table 9 in the Appendix. In all tasks, datasets were split into training, validation, and test sets in an 8:1:1 ratio. We applied two splitting methods: (1) In-Distribution Split, where the sets are randomly partitioned, and (2) Out-of-Distribution Split, where the sets are divided based on fingerprint similarity. This resulted in 26 tasks, allowing for a thorough evaluation of MRL models. The hyper-parameter search space is consistent across all tasks and baseline models (see Table 8 in the Appendix). Each set of hyper-parameters is trained 3 times using different random seeds, with the mean and standard deviation recorded. For all experiments, the checkpoint with the best validation loss is selected, and the corresponding test set results are reported.

## 4.2 Overall Results

Tables 1 and 2 present the overall comparison results for computational and experimental properties, respectively. The results clearly demonstrate SpaceFormer's superior performance. It ranks first in 22 out of 26 tasks and top two in 24 out of 26 tasks. SpaceFormer significantly outperforms all baselines in computational properties, with particularly strong results in the mu, R$^2$, and ZPVE tasks, where it surpasses the second-best models by an order of magnitude. Although it does not achieve the best results in a few experimental properties, the performance gap is minimal. In summary, by leveraging 3D space beyond atomic positions, SpaceFormer consistently outperforms previous MRL models across the comprehensive benchmark.

Table 2: Performance on molecular experimental property prediction tasks. The best results are highlighted in **bold**, and the second-best results are underlined.

| Model | HLM ↓ | MME ↓ | Solu ↓ | BBBP ↑ | BACE ↑ | HLM ↓ | MME ↓ | Solu ↓ | BBBP ↑ | BACE ↑ |
|---|---|---|---|---|---|---|---|---|---|---|
| | In-Distribution Split | | | | | Out-of-Distribution Split | | | | |
| GROVER | 0.4190 | 0.5362 | 0.4304 | 0.9210 | 0.9137 | 0.4667 | 0.4884 | 0.3466 | 0.8567 | 0.5084 |
| | ± 3.2e-2 | ± 4.9e-2 | ± 2.4e-2 | ± 8.5e-3 | ± 1.1e-2 | ± 2.2e-2 | ± 3.9e-2 | ± 3.8e-2 | ± 4e-2 | ± 2.8e-2 |
| GEM | 0.3013 | 0.3088 | 0.3511 | 0.9314 | 0.9406 | 0.3240 | 0.3110 | 0.3190 | 0.9024 | 0.6054 |
| | ± 7.9e-3 | ± 1.6e-3 | ± 4.8e-3 | ± 4.1e-3 | ± 3.6e-3 | ± 7.1e-3 | ± 2.2e-3 | ± 9e-3 | ± 2.3e-2 | ± 1.2e-2 |
| Uni-Mol | **0.2725** | 0.3033 | 0.3243 | 0.9397 | 0.9317 | 0.3026 | **0.2727** | 0.3295 | 0.8851 | **0.6793** |
| | ± 6.2e-3 | ± 1e-2 | ± 1.6e-2 | ± 1.1e-2 | ± 1.3e-2 | ± 6.5e-3 | ± 1.2e-2 | ± 1e-2 | ± 2e-2 | ± 3.3e-2 |
| Mol-AE | 0.2727 | 0.3000 | **0.3233** | 0.9366 | 0.9509 | 0.2843 | 0.2930 | 0.2983 | 0.9082 | 0.6406 |
| | ± 4.6e-3 | ± 6.7e-3 | ± 5.6e-3 | ± 6.5e-3 | ± 3.2e-3 | ± 4.7e-4 | ± 2e-2 | ± 2.3e-2 | ± 5.7e-2 | ± 2e-2 |
| SpaceFormer | 0.2774 | **0.2901** | 0.3320 | **0.9403** | **0.9523** | **0.2807** | 0.2794 | **0.2972** | **0.9099** | 0.6732 |
| | ± 3e-3 | ± 2.7e-3 | ± 1.1e-2 | ± 4.7e-3 | ± 7.6e-3 | ± 1.5e-3 | ± 3.2e-3 | ± 6.9e-3 | ± 2e-2 | ± 1.6e-2 |

Table 3: Ablation studies on PCA and in-cell position.

| No. | PCA | in-cell pos. | $R^2$ ↓ | ZPVE ↓ | $C_v$ ↓ | HOMO ↓ | pre-training cost |
|---|---|---|---|---|---|---|---|
| 1 | ✓ | ✓ | 2.8363 | 0.0003 | 0.0675 | 0.0017 | 32h |
| 2 | ✗ | ✓ | 3.3088 | 0.0004 | 0.0708 | 0.0018 | 35h |
| 3 | ✓ | ✗ | 5.8696 | 0.0004 | 0.0822 | 0.0020 | 32h |

## 4.3 ABLATION STUDIES

In this subsection, we conduct a series of experiments to evaluate the proposed components of SpaceFormer. We choose the $R^2$, ZPVE, $C_v$, and HOMO properties with the Out-of-Distribution Split for all ablation experiments.

**Gridding** As discussed in Sec 3.1, SpaceFormer integrates several techniques for efficient grid discretization while maintaining atomic precision. We focus on evaluating two key components: PCA for determining a minimal bounding cuboid for grid discretization, and in-cell positions to preserve atomic precision. The results, summarized in Table 3, lead to the following observations:

1. Impact of PCA: Comparing No. 1 and 2, we observe that omitting PCA significantly degrades the performance of SpaceFormer and slows training by approximately 10%. This suggests that PCA not only enhances model accuracy but also reduces training costs.

2. Impact of In-Cell Position: Comparing No. 1 and 3, we see that using in-cell positions leads to better performance. This demonstrates that incorporating in-cell positions effectively preserves atomic precision and contributes to superior performance.

**Grid Sampling** As discussed in Sec 3.2, we propose a sampling strategy for non-atom cells to further reduce training costs. In this ablation study, we conduct a series of experiments to evaluate the efficiency and performance of different sampling strategies. Specifically, we test various importance sampling strategies with different ratios ($m$) and temperatures ($\tau$), as well as several random sampling baselines. Additionally, we include two extreme cases: the atom-only model ($m = 0.0$) and the full-grid model ($m = 1.0$), to better assess the impact of non-atom cells. The results are summarized in Table 4, leading to the following conclusions:

1. Atom-Only Model (No. 1): This model performs the worst, demonstrating that non-atom cells, i.e., empty space, significantly contribute to improved model performance. This finding strongly supports the motivation behind our approach.

2. Full-Grid Model (No. 2): While this model shows strong performance, particularly for the HOMO property, its high computational cost renders it impractical for real-world applications.

3. Default Strategy (No. 4): The default sampling strategy used in SpaceFormer achieves the best balance between performance and efficiency. It is approximately 12 times faster than the full-grid model (No. 2), while delivering comparable performance.

4. Random Sampling Strategy (No. 6, 10, and 14-16): For random sampling, performance improves with a higher sampling ratio ($m$), but this also linearly increases training cost.

Table 4: Ablation studies on Grid Sampling. $m$ represents the sampling ratio of non-atom cells and $\tau$ represents the temperature for sampling. $\tau = $ '-' denotes the random sampling.

| No. | $m$ | $\tau$ | $R^2 \downarrow$ | ZPVE $\downarrow$ | $C_v \downarrow$ | HOMO $\downarrow$ | pre-training cost | avg. #cells |
|---|---|---|---|---|---|---|---|---|
| 1 | 0.0 | N/A | 3.5404 | 0.0004 | 0.0876 | 0.0025 | 12h | 0.1K |
| 2 | 1.0 | N/A | 2.8513 | 0.0004 | 0.0709 | 0.0015 | 389h | 6.3K |
| 3 | 0.1 | 0.5 | 2.8806 | 0.0004 | 0.0670 | 0.0017 | 32h | 0.8K |
| 4 | 0.1 | 1.0 | 2.8363 | 0.0003 | 0.0675 | 0.0017 | 32h | 0.8K |
| 5 | 0.1 | 2.0 | 3.0776 | 0.0005 | 0.0746 | 0.0017 | 32h | 0.8K |
| 6 | 0.1 | - | 2.8610 | 0.0005 | 0.0791 | 0.0018 | 32h | 0.8K |
| 7 | 0.2 | 0.5 | 2.9148 | 0.0004 | 0.0713 | 0.0017 | 51h | 1.4K |
| 8 | 0.2 | 1.0 | 3.2265 | 0.0004 | 0.0739 | 0.0016 | 51h | 1.4K |
| 9 | 0.2 | 2.0 | 3.2225 | 0.0004 | 0.0710 | 0.0016 | 51h | 1.4K |
| 10 | 0.2 | - | 2.8431 | 0.0004 | 0.0725 | 0.0017 | 51h | 1.4K |
| 11 | 0.4 | 0.5 | 3.2063 | 0.0007 | 0.0964 | 0.0017 | 103h | 2.6k |
| 12 | 0.4 | 1.0 | 3.6570 | 0.0006 | 0.0893 | 0.0018 | 103h | 2.6K |
| 13 | 0.4 | 2.0 | 2.9222 | 0.0004 | 0.0707 | 0.0016 | 103h | 2.6K |
| 14 | 0.4 | - | 2.9135 | 0.0004 | 0.0665 | 0.0016 | 103h | 2.6K |
| 15 | 0.6 | - | 2.9166 | 0.0004 | 0.0749 | 0.0017 | 193h | 3.9K |
| 16 | 0.8 | - | 2.8709 | 0.0003 | 0.0792 | 0.0016 | 300h | 5.1K |

Table 5: Ablation studies on 3D positional encoding.

| No. | 3D Directional PE with RoPE | 3D Distance PE with RFF | $R^2 \downarrow$ | ZPVE $\downarrow$ | $C_v \downarrow$ | HOMO $\downarrow$ |
|---|---|---|---|---|---|---|
| 1 | ✓ | ✓ | 2.8363 | 0.0003 | 0.0675 | 0.0017 |
| 2 | ✓ | ✗ | 3.4905 | 0.0004 | 0.0696 | 0.0017 |
| 3 | ✗ | ✗ | 3.7104 | 0.0004 | 0.1407 | 0.0022 |

5. Importance Sampling (No. 3–6 and No. 7–10): At smaller sampling ratios, the proposed importance sampling strategy based on eq.(4) outperforms random sampling, demonstrating its effectiveness in maintaining performance while improving efficiency.

6. However, at larger sampling ratios (No. 11-14), the importance sampling strategy cannot help to improve the performance. This is expected, as larger sampling ratios tend to include more grid cells near atoms, reducing the necessity of importance sampling strategy.

**3D Positional Encoding**   As detailed in Sec 3.3, we introduce two 3D positional encoding methods: 3D Directional Positional Encoding with RoPE (3D Directional PE with RoPE) and 3D Distance Positional Encoding with RFF (3D Distance PE with RFF). To evaluate their contributions to the final performance, we design two ablation models: one using only 3D Directional PE with RoPE (No. 2), and another excluding both proposed encodings (No. 3). In the latter, positional information is incorporated by simply adding the linear projection of the 3D position ($c_i$) to the input embeddings ($x_i$). The results in Table 5 clearly demonstrate that the proposed 3D positional encodings significantly enhance model performance.

## 4.4   IN-DEPTH COMPARISON WITH ATOM-BASED MRL MODELS

Previous experiments demonstrate the efficiency and effectiveness of SpaceFormer, but a key question remains: can incorporating empty space also enhance atom-based MRL models? To investigate this, we extend the strongest baseline, Uni-Mol, by incorporating randomly sampled empty points. Unlike SpaceFormer's grid discretization, the extended Uni-Mol samples points from continuous 3D space rather than grid cell centers. We evaluate various configurations with different numbers of sampled points/cells, as summarized in Table 6. For a fair comparison, we use the random sampling strategy for both Uni-Mol and SpaceFormer. The results lead to the following conclusions:

1. Baseline Comparison (No. 1 vs. No. 8): When excluding empty points/cells, the performance of Uni-Mol and SpaceFormer is comparable, indicating a similar baseline capability.

Table 6: Comparison with extended Uni-Mol using randomly sampled empty points. For fairness, SpaceFormer also uses random sampling strategy. As SpaceFormer samples a fraction of the entire grid, the number of sampled cells is not fixed, and the average number of sampled cells (marked with "*") is reported.

| No. | Model | # Empty Points | $R^2 \downarrow$ | ZPVE $\downarrow$ | $C_v \downarrow$ | HOMO $\downarrow$ | pre-training cost |
|---|---|---|---|---|---|---|---|
| 1 | Uni-Mol | 0 | 3.8530 | 0.0004 | 0.0914 | 0.0020 | 11h |
| 2 | Uni-Mol | 10 | 3.0506 | 0.0004 | 0.0820 | 0.0019 | 12h |
| 3 | Uni-Mol | 25 | 3.1586 | 0.0004 | 0.0886 | 0.0020 | 13h |
| 4 | Uni-Mol | 50 | 3.0141 | 0.0004 | 0.0973 | 0.0019 | 13h |
| 5 | Uni-Mol | 100 | 3.3509 | 0.0006 | 0.1114 | 0.0020 | 17h |
| 6 | Uni-Mol | 200 | 3.8193 | 0.0005 | 0.1145 | 0.0020 | 35h |
| 7 | Uni-Mol | 400 | 4.3522 | 0.0004 | 0.1337 | 0.0023 | 96h |
| 8 | SpaceFormer | 0 | 3.5404 | 0.0004 | 0.0876 | 0.0025 | 12h |
| 9 | SpaceFormer | 50* | 3.6770 | 0.0004 | 0.0805 | 0.0025 | 13h |
| 10 | SpaceFormer | 100* | 3.3996 | 0.0004 | 0.0777 | 0.0024 | 17h |
| 11 | SpaceFormer | 200* | 3.2388 | 0.0004 | 0.0787 | 0.0024 | 19h |
| 12 | SpaceFormer | 700* | 2.8610 | 0.0005 | 0.0791 | 0.0018 | 32h |
| 13 | SpaceFormer | 1300* | 2.8431 | 0.0004 | 0.0725 | 0.0017 | 51h |
| 14 | SpaceFormer | 1500* | 2.9135 | 0.0004 | 0.0665 | 0.0016 | 103h |

2. Impact of Empty Space on Uni-Mol: Incorporating a small number of empty points improves Uni-Mol's performance (No. 2 and 3 vs. No. 1), suggesting that even a limited representation of empty space can enhance the model performance.

3. Diminishing Returns for Uni-Mol: Increasing the number of empty points beyond a certain threshold does not yield further improvement (No. 3-7). This indicates that Uni-Mol struggles to utilize additional empty points effectively.

4. SpaceFormer's Scalability: In contrast, SpaceFormer continues to benefit from additional empty cells, with performance improving consistently as the number of empty cells increases (No. 8-14).

5. Efficiency of SpaceFormer: SpaceFormer scales much more efficiently with empty cells. While Uni-Mol's computational cost increases quadratically with more empty points (No. 1-7), Space-Former scales linearly (No. 8-14). For example, within 100 hours, SpaceFormer can process 1,500 cells, whereas Uni-Mol can only handle 400 points.

In summary, while incorporating empty space provides modest improvements to atom-based models like Uni-Mol, these gains are limited and come at a high computational cost. In contrast, Space-Former not only handles empty cells more efficiently but also achieves significantly better performance as the number of empty cells increases.

## 5 CONCLUSION

In this paper, we introduce SpaceFormer, a novel MRL framework that incorporates the 3D space beyond atomic positions to enhance molecular representation. To efficiently and effectively process 3D space, SpaceFormer leverages three key components: (1) Precision-Preserved Gridding, which discretizes continuous 3D space into a grid while maintaining atomic precision; (2) Grid Sampling, which improves efficiency by sampling grid cells without compromising accuracy; and (3) Linear-Complexity 3D Positional Encoding, which encodes pairwise positional information efficiently in 3D space. Extensive experiments validate the effectiveness and efficiency of SpaceFormer across various tasks.

Future research could explore two key areas: (1) investigating the theoretical foundations behind the effectiveness of incorporating empty space in MRL, as this work primarily provides empirical evidence, and (2) extending SpaceFormer to larger systems, such as proteins and complexes.

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

## A  EXPERIMENT DETAILS

The pretraining settings are detailed in Table 7, the downstream finetuning settings in Table 8, and the downstream tasks in Table 9.

Table 7: Pre-training Settings

| Hyper-parameters | Value |
|---|---|
| Peak learning rate | 1e-4 |
| LR scheduler | Linear |
| Warmup ratio | 0.01 |
| Total updates | 1M |
| Batch size | 128 |
| Residual dropout | 0.1 |
| Attention dropout | 0.1 |
| Embedding dropout | 0.1 |
| Encoder layers | 16 |
| Encoder attention heads | 8 |
| Encoder embedding dim | 512 |
| Encoder FFN dim | 2048 |
| MAE-Decoder layers | 4 |
| MAE-Decoder attention heads | 4 |
| MAE-Decoder embedding dim | 256 |
| MAE-Decoder FFN dim | 1024 |
| Adam $(\beta_1, \beta_2)$ | (0.9, 0.99) |
| Gradient clip | 1.0 |
| Mask ratio | 0.3 |
| Cell edge length $c_l$ | 0.49Å |
| $c_m$ for in-cell position discretization | 0.01Å |

Table 8: Fine-tuning Settings

| Hyper-parameters | Value |
|---|---|
| Peak learning rate | [5e-5, 1e-4] |
| Batch size | [32, 64] |
| Epochs | 200 |
| Pooler dropout | [0.0, 0.1] |
| Warmup ratio | 0.06 |

## B  ADDITIONAL COMPARISON WITH NEURAL POTENTIAL MODELS

As requested by the peer reviewers, we further compare SpaceFormer with neural potential models. Specifically, we use SchNet (Schütt et al., 2017) and PaiNN (Schütt et al., 2021) as baselines for benchmarking molecular property prediction and energy/force prediction tasks. All experiments are conducted using the same downstream hyperparameter settings described in Sec. 4.1.

For the molecular property prediction tasks, as presented in Table 10, SpaceFormer consistently outperforms both SchNet and PaiNN, demonstrating its superior predictive capabilities.

For the energy and force prediction tasks, evaluations were conducted on a subsampled version of QM7-X (Hoja et al., 2021). To investigate the models' few-shot learning capabilities, we randomly sampled training subsets containing 1k, 5k, 10k, and 20k samples, resulting in four separate experiments. All experiments utilized the same validation and test datasets, each containing 5k randomly sampled examples. All models were trained using energy loss only, with force errors computed as the gradients of the predicted energy with respect to atomic positions. The results, shown in Table 11, clearly demonstrate that SpaceFormer outperforms SchNet and PaiNN across all subset sizes.

Table 9: Summary information of the downstream datasets

| Category | Task | Task type | Metrics | # Samples | Describe |
|---|---|---|---|---|---|
| Computational Properties | mu | Regression | MAE | 40,000 | The measure of the molecule's permanent electric dipole moment |
| | alpha | Regression | MAE | 40,000 | The static polarizability of a molecule |
| | $R^2$ | Regression | MAE | 40,000 | The expectation value of the square of the electronic distance from the nucleus |
| | ZPVE | Regression | MAE | 40,000 | The energy associated with the vibrational motion of atoms in a molecule at absolute zero temperature. |
| | $C_v$ | Regression | MAE | 40,000 | The amount of heat needed to raise the temperature of a given amount of substance by one degree Celsius at constant volume |
| | HOMO | Regression | MAE | 40,000 | The highest energy molecular orbital that is occupied by electrons |
| | LUMO | Regression | MAE | 40,000 | The lowest energy molecular orbital that is not occupied by electrons |
| | GAP | Regression | MAE | 40,000 | The energy difference between the HOMO and LUMO |
| Experimental Properties | HLM | Regression | MAE | 3,087 | Human liver microsome stability |
| | MME | Regression | MAE | 2,642 | MDRR1-MDCK efflux ratio |
| | Solu | Regression | MAE | 2,713 | Aqueous solubility |
| | BBBP | Classification | AUC | 1,965 | Blood-brain barrier penetration |
| | BACE | Classification | AUC | 1,513 | Binding results of human BACE-1 inhibitors |

Table 10: Performance on molecular property prediction tasks with Out-of-Distribution split. The best results are highlighted in **bold**, and the second-best results are underlined.

| Model | mu ↓ (D) | alpha ↓ (Bohr$^3$) | $C_v$ ↓ (cal/(mol*K)) | HOMO ↓ (Hartree) | LUMO ↓ (Hartree) | GAP ↓ (Hartree) | HLM ↓ | MME ↓ | Solu ↓ |
|---|---|---|---|---|---|---|---|---|---|
| SchNet | 0.1554 ± 1.2e-3 | 0.1816 ± 2.6e-3 | 0.0671 ± 2.1e-3 | 0.0032 ± 4.3e-5 | 0.0028 ± 6.2e-5 | 0.0045 ± 8.8e-5 | 0.3863 ± 2.2e-2 | 0.3831 ± 2.2e-2 | 0.4419 ± 1e-2 |
| PaiNN | 0.0752 ± 1.8e-3 | 0.1518 ± 2.1e-2 | **0.0524** ± 1.6e-3 | 0.0028 ± 1.2e-5 | 0.0023 ± 7.5e-5 | 0.0040 ± 9.2e-5 | 0.3762 ± 6.8e-3 | 0.3539 ± 1.3e-2 | 0.4095 ± 1.9e-2 |
| SpaceFormer | **0.0493** ± 1.3e-3 | **0.1425** ± 3.1e-3 | 0.0675 ± 1.4e-3 | **0.0017** ± 1.3e-5 | **0.0019** ± 3.3e-5 | **0.0031** ± 3.1e-5 | **0.2807** ± 1.5e-3 | **0.2794** ± 3.2e-3 | **0.2972** ± 6.9e-3 |

## C   DETAILS ABOUT 3D DIRECTIONAL POSITIONAL ENCODING WITH RoPE

To make it easier to understand, we have added more details of 3D Directional Positional Encoding with RoPE here.

In 1D case, such as natural language processing, given two tokens located at positions $x_i$ and $x_j$, the original RoPE mechanism is designed to capture their relative position $x_j - x_i$. This concept is straightforward and widely accepted.

Next, we extend this concept to 2D. Given two points in 2D space with positions $(x_i, y_i)$ and $(x_j, y_j)$, the goal is to encode their positional differences $x_j - x_i$ and $y_j - y_i$. 2D RoPE achieves this by encoding each positional difference independently. Specifically, in the context of multi-head

Table 11: Performance on molecular energy prediction tasks with QM7-X dataset. The best results are highlighted in **bold**, and the second-best results are underlined.

| Model | # Training Samples | Energy MAE (eV) ↓ | Force MAE (eV/Å) ↓ |
|---|---|---|---|
| SchNet | 1k | $0.5403 \pm 0.0103$ | $0.8995 \pm 0.0191$ |
| PaiNN | 1k | $\underline{0.4064} \pm 0.0061$ | $\underline{0.7425} \pm 0.0213$ |
| SpaceFormer | 1k | $\mathbf{0.3841} \pm 0.0310$ | $\mathbf{0.5990} \pm 0.0253$ |
| SchNet | 5k | $0.2459 \pm 0.0013$ | $0.5683 \pm 0.0086$ |
| PaiNN | 5k | $\underline{0.1839} \pm 0.0016$ | $\underline{0.4166} \pm 0.0042$ |
| SpaceFormer | 5k | $\mathbf{0.1449} \pm 0.0015$ | $\mathbf{0.3025} \pm 0.0004$ |
| SchNet | 10k | $0.1825 \pm 0.0039$ | $0.4494 \pm 0.0089$ |
| PaiNN | 10k | $\underline{0.1444} \pm 0.0015$ | $\underline{0.3386} \pm 0.0036$ |
| SpaceFormer | 10k | $\mathbf{0.1061} \pm 0.0004$ | $\mathbf{0.2360} \pm 0.0008$ |
| SchNet | 20k | $0.1435 \pm 0.0018$ | $0.3627 \pm 0.0059$ |
| PaiNN | 20k | $\underline{0.1057} \pm 0.0013$ | $\underline{0.2602} \pm 0.0015$ |
| SpaceFormer | 20k | $\mathbf{0.0789} \pm 0.0006$ | $\mathbf{0.1829} \pm 0.0013$ |

attention, half of the attention heads are assigned to encode $x_j - x_i$, and the other half to encode $y_j - y_i$.

Similarly, this concept extends naturally to 3D: we encode the relative positional differences along all three axes $(x_j - x_i, y_j - y_i, z_j - z_i)$ by dividing the attention heads into three sets. Each set is dedicated to encoding the positional difference along one axis. This ensures that 3D RoPE directly encodes relative positions in 3D space, without involving any 2D projections or rotations.

From the above explanation, it is clear that the key to 3D RoPE is using three independent sets of 1D RoPE to encode relative positions along the three axes.

# D  ADDITIONAL ABLATION STUDIES ON PCA

To demonstrate the contribution of PCA to the final performance, we show additional ablation experimental results in Table 12.

From the results, it is clear that PCA does not significantly contribute to the final performance, while the proposed positional encoding techniques (RoPE and RFF) play a much larger role in improving the final performance.

The results clearly demonstrate that SpaceFormer achieves strong performance even without PCA and under randomly rotated 3D inputs. In our implementation, a unique random rotation is applied at each epoch, exposing the model to a broader variety of orientations over time. This enhances its ability to generalize across different coordinate systems and confirms that the performance improvement is not due to PCA. Instead, the model effectively learns arbitrary 3D directions rather than relying on "memorizing geometries in PCA frames."

Specifically, random rotations expose the model to a diverse range of coordinate systems with varying orientations, enabling more comprehensive learning. Without random rotations, certain coordinate systems may be underrepresented during training, potentially degrading performance during inference on less common orientations. Random rotations ensure a more uniform distribution of coordinate systems, thereby improving robustness. In the main text, we primarily use PCA instead of random rotations to reduce the molecular space requiring processing and to enhance computational efficiency.

Table 12: Ablation studies on PCA and random rotation.

| No. | PCA | RoPE | RFF | Random Rotation | $R^2 \downarrow$ | ZPVE $\downarrow$ | $C_V \downarrow$ | HOMO $\downarrow$ |
|---|---|---|---|---|---|---|---|---|
| 1 | ✓ | ✓ | ✓ | ✗ | 2.8363 | 0.00028366 | 0.0675 | 0.001687503 |
| 2 | ✗ | ✓ | ✓ | ✗ | 3.3088 | 0.00040852 | 0.0708 | 0.00175726 |
| 3 | ✓ | ✗ | ✗ | ✗ | 3.7104 | 0.000449727 | 0.1407 | 0.002166273 |
| 4 | ✗ | ✓ | ✓ | ✓ | 2.9840 | 0.00032006 | 0.0674 | 0.00147432 |

