# OpenReview forum: "More Space Is All You Need: Revisiting  Molecular Representation Learning"
_ICLR.cc/2025/Conference — ICLR 2025 Conference Withdrawn Submission_

### Official Review · Reviewer_jmPG · 2024-10-17

**Soundness:** 2
**Presentation:** 3
**Contribution:** 2
**Rating:** 5
**Confidence:** 4

**Summary:**

This paper proposes SpaceFormer, a transformer-based model for Molecular Representation Learning (MRL) that incorporates information from the 3D space beyond atomic positions. The authors argue that this empty space contains valuable information relevant to molecular properties. SpaceFormer receives as input the discretized 3D space around a molecule as a grid of cells, where cells containing atoms are distinguished from non-atom cells. For better computational efficiency, the non-atom cells are subsampled using an importance sampling strategy. The authors conduct experiments to assess the performance of SpaceFormer on several benchmarks for molecular property prediction.

**Strengths:**

+ The authors perform ablation studies to analyze the contribution of each component of SpaceFormer, providing insights into the importance of each design choice. The ablation studies on grid sampling are particularly informative, showcasing the trade-off between performance and efficiency.

+ The authors extend the Uni-Mol baseline by incorporating randomly sampled empty points, allowing for a direct comparison between an existing atom-based models with and without empty space consideration. This strengthens the claim that SpaceFormer's approach of explicitly modeling empty space within a grid structure is more effective.

**Weaknesses:**

- The benchmark results for QM9 shown in Table 1 suggest that the performance of SpaceFormer would be SOTA, but this is not the case. Several published atom-based models designed for regression on molecular properties achieve significantly better results (see e.g. http://proceedings.mlr.press/v139/schutt21a/schutt21a.pdf for an already 3 year old model that is better). Further, the authors should explicitly report the units (I simply assumed that the default QM9 units are used), otherwise no meaningful comparisons to published work are possible.

- While the paper reports performance on QM9, this is a quite old and rather simple benchmark (perhaps comparable to MNIST in computer vision) and the field has since moved to more sophisticated benchmarks, e.g. MD17, MD22, or QM7-X. I think additional experiments and comparisons to SOTA models on (at least one of) these datasets would be very insightful.

- The ablation studies and the comparison to an atom-only model is a good start, but I believe they are flawed: Since the atom-only model processes significantly fewer tokens, it has in a sense "less internal FLOPs" compared to models that also process tokens for empty space. It would be more insightful to see results for an experiment where this is controlled for, for example by increasing the width/depth of the atom-only model to achieve "compute-parity". This would allow to see whether the empty space tokens are useful conceptually, or whether they simply increase the amount of compute the model can leverage.

- The resemblance between electron density and the learned representations shown in Figure 2, as well as the conclusions drawn from it (e.g. that this is evidence that SpaceFormer learns meaningful physical relationships) is questionable. It is trivial to produce a surface that resembles electron density simply by using the distance to the atoms. For example, just take the $\sum_{i=1}^{N} \lVert \vec{r} - \vec{r}_i \rVert^{-2} = 1 a_0^{-2}$ isosurface and you will get something that resembles the electron density much more closely than the representations learned by SpaceFormer. (Here, the sum runs over the $N$ atoms in the structure, $\vec{r}_i$ is the position of the $i$-th atom, $\vec{r}$ is the "query point" in 3D space, and $a_0$ means Bohr, i.e., the atomic unit of length.)

- While the authors motivate incorporating empty space by mentioning some physical principles/facts (e.g. that electric fields and electron density extend beyond atoms), the paper lacks a formal theoretical analysis of why and how this information contributes to better molecular representations. A theoretical foundation would significantly strengthen the paper's contributions.

**Questions:**

* Can you provide a quantitative comparison between the learned representations and the corresponding electron density maps? As a baseline, something like a distance-based isosurface (see above) would be useful (the used isovalue should of course be optimized, I just put $1 a_0^{-2}$ for simplicity).

* How do you choose the coordinate system in cases where the eigenvectors determined by PCA are degenerate (for example for molecules with $\mathrm{T_d}$ or $\mathrm{O_h}$ symmetry)?

* The authors mention that the encoding of angular directions may be unstable when the entire system undergoes global rotations. I don't see how this can happen if you choose a coordinate system based on PCA eigenvectors (this should give a sort of "canonical orientation"). Can the authors elaborate?

* How does the performance of SpaceFormer vary with different grid resolutions? Have you experimented with adaptive gridding strategies to address potential limitations for larger molecules or systems with varying densities?

* What are the specific challenges you anticipate in extending SpaceFormer to larger systems like proteins, and what potential mitigation strategies are you considering?

**Additional Feedback:**

* Consider adding a discussion about the computational cost of SpaceFormer during training and inference, compared to other 3D MRL models.

* The paper mentions future work on extending SpaceFormer to larger systems like proteins. However, the current grid-based approach could face scalability issues with larger systems due to the cubic growth in the number of cells. Discussing potential strategies for mitigating these challenges would be helpful.

---

> ### Author Response · Authors · 2024-11-18
> **response to Reviewer jmPG (1/2)**
>
> We thank the reviewer for their thoughtful comments and concerns, particularly regarding the experimental results, including those on QM9. Below, we address these points in detail:
>
> 1. **On the performance of QM9**:
>
> It is important to note that the QM9 dataset used in this paper is a **subsampled** version containing **40k samples**, whereas the original QM9 dataset consists of **133k samples**. Therefore, direct comparison of our results with those reported in previous papers using the full dataset would be unfair. The subsampling in our study is intentional, as the paper focuses on evaluating the performance of a pretrained MRL model on downstream tasks with limited data. In real-world applications, the availability of labeled data is often constrained due to the high costs of computation and wet-lab experiments. This setup, therefore, enables a more realistic assessment of the model’s effectiveness in practical scenarios. Regarding the units in QM9, we use the original units provided in the dataset. We will revise the paper to explicitly include this detail.
>
> 2. **On more benchmark datesets than QM9**:
>
> We would like to clarify that our benchmark datasets are not limited to QM9. In addition to QM9, we include tasks from MoleculeNet (2018) and Biogen ADME (2023). These datasets focus on wet-lab experimental properties, most of which have very limited data, making them well-suited for evaluating pretrained MRL models on real-world tasks.
> The reviewer’s suggested datasets -- MD17, MD22, and QM7-X -- are not ideal for evaluating pretrained MRL models in our scenario. Specifically:
>
> - MD17 and MD22: These datasets focus on molecular dynamics and are primarily used to train neural potential models for predicting energy and forces, which differ from the property prediction tasks targeted by our method.
> - QM7-X: While QM7-X provides a large number of labeled samples, it does not align with the primary scenario of our work, which focuses on downstream tasks with limited data.
>
> Although these datasets are not suitable for evaluation in our current framework, we agree that they could serve as valuable datasets for pretraining MRL models. We plan to investigate their utility for this purpose in future work.
>
> 3. **On the Claim That the Comparison with Atom-Based Models Is “Flawed”**:
>
> We regret that we do not fully understand the reviewer’s assertion that our comparison with atom-based models is “flawed.” As shown in Table 6, we explicitly incorporate **additional empty points** into the atom-based model. The table clearly demonstrates the increased computation times as more empty points are added, reflecting a significant increase in “internal FLOPS” for Uni-Mol.
> The primary limitation here is that Uni-Mol is not efficient when scaling with a large number of empty points, which is why we restricted our experiments to 400 empty points. However, the observed trend suggests that even with more empty points, the performance does not improve. This finding underscores the inherent limitations of atom-based models in leveraging empty space information effectively.
>
> 4. **On the Quantitative Comparison with the Electron Density Map**:
>
> We thank the reviewer for pointing this out and conducted a further comparison with the electron density map. Specifically, we directly compared the normalized density map from the learned representation with the electron density by calculating the element-wise absolute error at grid cells and averaging the results over 50 data samples. The resulting error is approximately 7.4e-4. We also implemented the distance-based solution suggested by the reviewer, which yielded an error of about 7.9e-4.
> While the learned representation slightly outperforms the distance-based solution, the improvement is small. We agree that this comparison may not be particularly meaningful at this stage due to the small gain. As a result, we will remove this analysis from the paper and plan to conduct a deeper investigation into the learned representation in future work. We sincerely thank the reviewer for this suggestion, as it has helped us improve the quality of the paper.

---

> > ### Comment · Reviewer_jmPG · 2024-11-18
> >
> > The authors try to avoid including additional benchmarks by stating they would not be "suitable for evaluation in our current framework". However, this is just an arbitrary decision by the authors, they give no proper arguments why additional benchmarks cannot be run (it seems the authors simply do not want to do this). I therefore have to assume that their proposed model does not perform well in the suggested evaluations.
> >
> > My other points (with the exception of my critique of the electron density comparison, which seems to have convinced the authors) are addressed in a similar manner (the authors avoid to address my criticism and instead evade or seem to purposefully misunderstand my points). I therefore feel that further discussion is unproductive and remain at my present score.

---

> > > ### Author Response · Authors · 2024-11-18
> > >
> > > We respectfully disagree with the assertion that this is “an arbitrary decision.” The benchmarks suggested by the reviewer, such as MD17 and MD22, are specifically designed for **neural potential models**, which focus on energy and force prediction tasks, such as molecular dynamics simulations. These benchmarks aim to evaluate models that approximate potential energy surfaces with high accuracy for quantum chemistry and molecular mechanics applications.
> > >
> > > In contrast, our paper focuses on **molecular representation learning** (MRL), where the primary goal is to pretrain on large-scale unlabeled molecular data and transfer the learned representations to downstream tasks, such as property prediction, with limited labeled data. While there are conceptual overlaps, MRL models and neural potential models target fundamentally different applications and require distinct evaluation protocols.
> > >
> > > Furthermore, most MRL studies, including the baselines referenced in our manuscript, do not utilize benchmarks like MD17 or MD22, as they are not aligned with the objectives of MRL.
> > >
> > > If the reviewer deems it necessary, we are willing to subsample a subset from QM7-X and conduct an additional benchmark analysis.

---

> > > > ### Comment · Reviewer_jmPG · 2024-11-18
> > > >
> > > > It is unclear to me why the authors think energy prediction is not a valid downstream task. The energy of a molecule *is* a property after all (arguably one of the most relevant ones), and good molecular representations are a prerequisite to solve this task well. If the primary goal is to pretrain on large-scale unlabeled molecular data and transfer the learned representations to downstream tasks, energy prediction seems like an ideal test to me to. If the learned molecular representations are useful in general, this task should pose no issue and would be a very valuable addition. If the representations do not perform well on this task, this means that the learned molecular representations have limitations which need to be investigated and discussed.
> > > >
> > > > And yes, I do believe that this is necessary to be able to properly evaluate the proposed method (however, it is not important whether this test is done on MD17, QM7-X, or some other established benchmark).

---

> > > > > ### Author Response · Authors · 2024-11-18
> > > > >
> > > > > Thanks the reviewer very much for providing further clarification regarding the task in MD17/MD22. If the goal is specifically energy prediction, we will conduct the additional benchmark and update the results here as soon as they are available.

---

> > > > > > ### Comment · Reviewer_jmPG · 2024-11-18
> > > > > >
> > > > > > For further clarity: Energy *and* force prediction, as only the forces will reveal whether the learned molecular representations are sufficiently smooth under molecular deformations.

---

> > > > ### Author Response · Authors · 2024-11-18
> > > >
> > > > For electron density comparison, we have conducted the additional quantitative analysis as suggested by the reviewer and found that our approach achieves slightly better results than the distance-based method (7.4e-4 vs. 7.9e-4). However, given the small gap, we acknowledge the reviewer’s criticism and agree that this visualization may not provide meaningful insights. As a result, we will remove it from the paper. If we have misunderstood the reviewer’s point, we kindly ask for clarification, and we are more than willing to address any further concerns.

---

> ### Author Response · Authors · 2024-11-18
> **response to Reviewer jmPG (2/2)**
>
> 5. **On PCA and SE(3) Invariance**:
>
> PCA is not intended to handle SE(3) invariance in our method. Its motivation is solely to reduce the grid size, not to address SE(3) invariance. Also, as reviewer suggested, in molecular data, the inherent symmetry often causes PCA to fail to produce a unique coordinate system, thereby making it unreliable for ensuring SE(3) invariance. As a result, even with PCA, we still need a stable method to encode SE(3)-invariance features, like pair-wise distances.
> To handle cases where PCA eigenvectors are degenerate, we use atom weights to break the symmetry by determining the axis directions based on the side with the larger mean atom weight. Additionally, we have a fallback mechanism: if PCA fails to produce a valid coordinate system (e.g., non-orthogonality or axis swapping), we revert to the original coordinate system to ensure robustness.
>
> 6. **On adaptive gridding strategies**:
>
> Thank you for the suggestion. In the early stages of our work, we experimented with an adaptive gridding strategy based on an octree structure. Specifically, eight adjacent empty cells could be merged in a bottom-up, hierarchical manner, resulting in fewer but larger cells in coarse regions and more but smaller cells in fine regions with higher atomic density. This approach is conceptually similar to our proposed importance sampling strategy, which samples fewer cells in coarse regions and more cells in fine regions.
> However, the proposed sampling strategy offers better control over the number of sampled cells, providing greater flexibility and efficiency. For this reason, we adopted it in the final design.
>
> 7. **On extending SpaceFormer to larger systems**:
>
> We are actively working on extending SpaceFormer to handle larger systems. With the proposed grid sampling strategy, the total number of cells can be effectively controlled, but two additional bottlenecks remain:
>
> - Full-Length Attention: As the number of cells increases, the current full-length attention mechanism becomes a significant bottleneck. To address this, we plan to adopt locality-based optimizations such as sliding-window attention or Swin attention. Furthermore, incorporating local-to-global or multi-level hierarchical attention mechanisms could further enhance both scalability and performance.
>
> - Extremely Large Systems: While the above optimizations allow SpaceFormer to handle most cases in protein databases, a small number of extremely large proteins still pose challenges. For these cases, we utilize a random spatial cropping strategy during training to reduce the computational load. Notably, cropping is disabled during inference to ensure accurate predictions.
>
> 8. **On discussion about the computational cost of SpaceFormer and other models**:
>
> We discuss the computational cost of SpaceFormer in Section 4.4, Point 5. Specifically, we note that "SpaceFormer scales much more efficiently with empty cells. While Uni-Mol’s computational cost increases quadratically with the number of empty points (No. 1-7), SpaceFormer scales linearly (No. 8-14). For example, within 100 hours, SpaceFormer can process 1,500 cells, whereas Uni-Mol can only handle 400 points."
> If this discussion is not enough, we would appreciate more details on specific aspects of computational cost that we can further elaborate on.

---

> ### Author Response · Authors · 2024-11-18
>
> Thank you very much for the further clarification. As our model does not support force prediction for now, we may not be able to complete it within the discussion period. However, we will ensure the energy prediction benchmark is completed and will make our best effort to include force prediction as well.

---

> > ### Comment · Reviewer_jmPG · 2024-11-18
> >
> > I thank the authors for their efforts. Perhaps it is helpful to note that the forces are just the negative gradient of the energy with respect to the atomic positions, which typically can be implemented with one line of code in automatic differentiation frameworks  such as pytorch or jax. In case the authors use an unusual setup that has no equivalent of jax.grad or torch.backward, the forces can still be easily estimated via numerical differentiation using a finite difference scheme (this requires at least 3N energy evaluations for a molecule with N atoms, but should be affordable for small molecules such as those in QM7-X). While numerical differentiation scheme is not ideal, it should still be sufficient to determine whether the learned molecular representations vary sufficiently smoothy when the structure is distorted.

---

> ### Author Response · Authors · 2024-11-18
>
> We sincerely thank the reviewer for the kind suggestion. Based on our current understanding of neural potential networks, gradient-based force prediction typically requires a supervised loss on forces during training. This process involves higher-order gradients, which may present challenges in our current framework (e.g., the flash-attention kernel might not support it). Nevertheless, we will make every effort to implement it.

---

> > ### Comment · Reviewer_jmPG · 2024-11-18
> >
> > Yes, this is correct if you want to include forces in your loss function. However, to check that the learned molecular representations are smooth, it is sufficient to train on energies only (the forces can be calculated only at test time if a force loss turns out to be problematic with your setup). At least for the MD17 benchmark, I know that there are also published results for force predictions when training on energies only to which you could compare to if necessary.

---

> > > ### Author Response · Authors · 2024-11-23
> > > **New results on energy and force prediction**
> > >
> > > We have conducted the additional experiment (energy and force prediction) requested by the reviewer, and the results are presented in the table below. These results clearly demonstrate that our model outperforms both SchNet and PaiNN. Completing these additional experiments required significant time and effort, underscoring our commitment to thoroughly addressing the reviewer’s concerns. We kindly request the reviewer to recognize our dedication and reconsider their evaluation.
> > >
> > > Experimental Setup:
> > >
> > > - Dataset: We used QM7-X for this evaluation. To assess the few-shot learning capabilities of the models, we randomly sampled training subsets containing 1k, 5k, 10k, and 20k samples, resulting in four separate experiments. All experiments share the same validation and test datasets, each consisting of 5k randomly sampled samples.
> > > - Baseline Models: We compared our model with SchNet and PaiNN, implemented using SchNetPack (https://github.com/atomistic-machine-learning/schnetpack). All models were **trained using energy loss only**. The force errors were calculated as the gradients of the energy with respect to atomic positions.
> > > - Hyperparameters: All experiments used the same hyperparameter search space (described in Table 8). Each hyperparameter configuration was trained three times with different random seeds, and we report the mean and standard deviation of the results. For all models, the checkpoint with the best validation loss was selected, and the corresponding test set results are reported.
> > >
> > >
> > > | Model          | # Training Samples | Energy MAE (eV) ↓      | Force MAE (eV/Å) ↓      |
> > > |-----------------|--------------------|-------------------------|-------------------------|
> > > | **SchNet**     | 1k                | 185.1806 ± 6.6565       | 146.4527 ± 5.8946       |
> > > | **PaiNN**      | 1k                | 167.6489 ± 30.9216      | 153.2552 ± 19.6824      |
> > > | **SpaceFormer**| 1k                | **36.9473** ± 7.0779        | **28.6776** ± 2.9315        |
> > > | **SchNet**     | 5k                | 74.5823 ± 4.9014        | 88.3705 ± 3.5321        |
> > > | **PaiNN**      | 5k                | 24.8025 ± 1.9379        | 29.7473 ± 2.4853        |
> > > | **SpaceFormer**| 5k                | **8.3037** ± 1.6529         | **9.5590** ± 2.5805         |
> > > | **SchNet**     | 10k               | 47.7015 ± 1.1859        | 65.1444 ± 2.4222        |
> > > | **PaiNN**      | 10k               | 15.6893 ± 0.7267        | 21.7204 ± 1.2478        |
> > > | **SpaceFormer**| 10k               | **4.3718** ± 0.6450         | **6.8527** ± 1.5114         |
> > > | **SchNet**     | 20k               | 31.2351 ± 1.0555        | 44.2606 ± 0.5122        |
> > > | **PaiNN**      | 20k               | 8.8318 ± 1.4907         | 12.3451 ± 1.7097        |
> > > | **SpaceFormer**| 20k               | **2.0994** ± 0.1833         | **3.3146** ± 0.3387         |

---

> > > > ### Comment · Reviewer_jmPG · 2024-11-25
> > > >
> > > > I sincerely thank the authors for running these additional experiments.
> > > >
> > > > Can the authors please double-check that the reported units are really in eV and eV/Å? I ask because the reported errors seem much too high for these units. In fact, they are so high that there is almost certainly a bug (or the units are wrong).
> > > >
> > > > To get a rough feeling for the magnitude of sensible errors on QM7-X, I fitted a linear model on a very simple molecular descriptor that just counts the number of H, C, N, O, S, and Cl atoms in each structure (so e.g. CH4 would have the descriptor [4, 1, 0, 0, 0, 0]). Since this model completely ignores the atomic positions, it always predicts zero forces. Even with this very simple baseline (the model has only 6 parameters), I obtain MAEs of ~2.2 eV for energies and ~1.2 eV/Å for forces on QM7-X.
> > > >
> > > > Energy and force errors reported in the literature for the SchNet and PaiNN models are typically well below 0.1 eV(/Å). Admittedly, when training on energies only, errors are usually a bit higher (see e.g. https://arxiv.org/abs/2007.09593), but 2-3 orders of magnitude larger errors seem unreasonable to me (and also do not align with my own experience when training these models on energies only vs. energy + forces).

---

> > > > > ### Author Response · Authors · 2024-11-25
> > > > >
> > > > > Thank you for your detailed comment. After reviewing the configuration, we identified the issue: the per-atom reference energy was not utilized in the experiments mentioned above.
> > > > >
> > > > > Specifically, since SchNetPack does not provide a pre-defined training configuration for QM7-X, we adopted the MD17 configuration (https://github.com/atomistic-machine-learning/schnetpack/blob/master/src/schnetpack/configs/experiment/md17.yaml), where the hyperparameter add_atomrefs is set to False by default. With this setting, the model directly fits the total energy without subtracting the per-atom reference energy, which led to degraded performance.
> > > > >
> > > > > In SpaceFormer, the per-atom reference energy is also not used, so the comparison remains fair.
> > > > >
> > > > > We are currently addressing this issue by reconfiguring and rerunning the experiments. We sincerely apologize for this oversight, as this was our first experience using SchNetPack. Thank you for pointing it out.

---

> > > > > > ### Comment · Reviewer_jmPG · 2024-11-25
> > > > > >
> > > > > > No worries, glad to hear you could identify the bug so quickly! I'm looking forward to the updated results.

---

> > > > > > > ### Author Response · Authors · 2024-11-26
> > > > > > >
> > > > > > > We thank the reviewer for their patience. The experiment with atom reference energy has been completed, and the results are presented in the table below:
> > > > > > >
> > > > > > > | Model          | # Training Samples | Energy MAE (eV) ↓      | Force MAE (eV/Å) ↓      |
> > > > > > > |-----------------|--------------------|-------------------------|-------------------------|
> > > > > > > | **SchNet**     | 1k                | 0.5403 ± 0.0103       | 0.8995 ± 0.0191       |
> > > > > > > | **PaiNN**      | 1k                | 0.4064 ± 0.0061      | 0.7425 ± 0.0213      |
> > > > > > > | **SpaceFormer**| 1k                | **0.3841** ± 0.0310        | **0.5990** ± 0.0253        |
> > > > > > > | **SchNet**     | 5k                | 0.2459 ± 0.0013       | 0.5683 ± 0.0086        |
> > > > > > > | **PaiNN**      | 5k                |   0.1839 ± 0.0016      | 0.4166 ± 0.0042        |
> > > > > > > | **SpaceFormer**| 5k                | **0.1449** ± 0.0015         | **0.3025** ± 0.0004         |
> > > > > > > | **SchNet**     | 10k               | 0.1825 ± 0.0039        | 0.4494 ± 0.0089        |
> > > > > > > | **PaiNN**      | 10k               | 0.1444 ± 0.0015        | 0.3386 ± 0.0036        |
> > > > > > > | **SpaceFormer**| 10k               | **0.1061** ± 0.0004         | **0.2360** ± 0.0008         |
> > > > > > > | **SchNet**     | 20k               | 0.1435 ± 0.0018        | 0.3627 ± 0.0059        |
> > > > > > > | **PaiNN**      | 20k               | 0.1057 ± 0.0013         | 0.2602 ± 0.0015        |
> > > > > > > | **SpaceFormer**| 20k               | **0.0789** ± 0.0006         | **0.1829** ± 0.0013         |

---

> > > > > > > > ### Comment · Reviewer_jmPG · 2024-11-26
> > > > > > > >
> > > > > > > > I sincerely thank the authors for adding these experiments, I think they are absolutely necessary for evaluating the proposed method (although I am extremely surprised by the bad performance of SchNet and PaiNN, which in my experience should achieve much lower errors when trained on 20k structures, even without including forces in the loss).
> > > > > > > >
> > > > > > > > Since the authors have demonstrated their willingness to improve the paper, I am raising my score. However, I agree with Reviewer rJDR in that some important points are still not addressed fully (see also my initial review), so I cannot raise my score any higher.

---

> > > > > > > > > ### Author Response · Authors · 2024-11-26
> > > > > > > > >
> > > > > > > > > We are less familiar with the energy and force prediction task, but we suspect the reason for the increased difficulty lies in the nature of the QM7-x dataset, which contains multiple molecules. This is in contrast to tasks like MD17, which typically involve only single molecules. The diversity in QM7-x likely increases the learning challenge.
> > > > > > > > >
> > > > > > > > > We sincerely thank the reviewer for the increased score. Could you kindly clarify which “important points” remain unaddressed? We are more than willing to further address them.

---

> > > > > > > > > > ### Comment · Reviewer_jmPG · 2024-11-27
> > > > > > > > > >
> > > > > > > > > > As mentioned in my original reply, it feels like some of the points raised have been evaded.
> > > > > > > > > >
> > > > > > > > > > For example, I pointed out that the table for QM9 lacks comparisons to more recent models, which achieve better results. The authors tried to argue that they cannot compare to these numbers, since they purposefully subsampled to 40k structures, while SOTA models are trained on more structures. Why not simply train their model on the same number of structures to be able to do the comparison? This seems like the authors want to avoid comparison to SOTA models.
> > > > > > > > > >
> > > > > > > > > > Similarly, I pointed out that the comparison to the atom-based model is flawed, because the authors should do the comparison under an "iso-FLOP setting". The authors argue that they are limited by Uni-Mol's non-efficiency when scaling to a larger number of points, and that the observed trends suggests that even with more empty points, performance would not improve. However, this is speculation, the experiment should actually be run (under an iso-FLOP constraint). I understand that it might be hard to scale up Uni-Mol to do a fair comparison, but the authors could simply scale their model down instead. Again, it feels like this point as been purposefully evaded with (in my opinion) quite weak arguments.
> > > > > > > > > >
> > > > > > > > > > The above points only relate to my own review, but I also agree with the other reviewers, who also feel some of their points have not been fully addressed. Please refer to their reviews and replies.

---

> > > > > > > > > > > ### Author Response · Authors · 2024-11-27
> > > > > > > > > > >
> > > > > > > > > > > We sincerely thank the reviewer for their additional comments. We are happy to address them further.
> > > > > > > > > > >
> > > > > > > > > > > ---
> > > > > > > > > > >
> > > > > > > > > > > ### 1. Performance on QM9
> > > > > > > > > > >
> > > > > > > > > > > To ensure a fair comparison, we conducted an additional experiment, as noted in our earlier reply ([link to discussion](https://openreview.net/forum?id=LBsr2llHz0&noteId=AL0f3opV3m)). We apologize for not including the results in this thread. Specifically, using the SchNetPack code, we trained SchNet and PaiNN on our 40k downstream dataset. The results are summarized below:
> > > > > > > > > > >
> > > > > > > > > > > | Model          | QM9_mu ↓        | QM9_HOMO ↓      | QM9_LUMO ↓      | QM9_GAP ↓       | HLM ↓             | MME ↓             | Solu ↓           |
> > > > > > > > > > > |-----------------|-----------------|-----------------|-----------------|-----------------|-------------------|-------------------|------------------|
> > > > > > > > > > > | **SchNet**      | 0.1554 ± 1.2e-3 | 0.0032 ± 4.3e-5 | 0.0028 ± 6.2e-5 | 0.0045 ± 8.8e-5 | 0.3863 ± 2.2e-2  | 0.3831 ± 2.2e-2  | 0.4419 ± 1e-2    |
> > > > > > > > > > > | **PaiNN**       | _0.0752_ ± 1.8e-3 | _0.0028_ ± 1.2e-5 | _0.0023_ ± 7.5e-5 | _0.0040_ ± 9.2e-5 | _0.3762_ ± 6.8e-3 | _0.3539_ ± 1.3e-2 | _0.4095_ ± 1.9e-2 |
> > > > > > > > > > > | **SpaceFormer** | **0.0493** ± 1.3e-3  | **0.0017** ± 1.3e-5 | **0.0019** ± 3.3e-5 | **0.0031** ± 3.1e-5 | **0.2807** ± 1.5e-3 | **0.2794** ± 3.2e-3 | **0.2972** ± 6.9e-3 |
> > > > > > > > > > >
> > > > > > > > > > > ---
> > > > > > > > > > >
> > > > > > > > > > > ### 2. Comparison with Uni-Mol
> > > > > > > > > > >
> > > > > > > > > > > We would like to clarify that we did not "purposefully evade" this comparison. The primary limitation lies in the $O(N^2)$ memory cost of Uni-Mol, which makes it impractical to train with a very large number of empty points. Even if it were feasible, it would be prohibitively expensive, requiring significant resources such as A100/H100 GPUs and considerable time. For example, training Uni-Mol with 400 empty points already takes 4 days.
> > > > > > > > > > >
> > > > > > > > > > > However, we have provided a fair comparison by scaling SpaceFormer to match Uni-Mol's computational costs, as shown in Table 6 of the paper. Below, we summarize the key findings:
> > > > > > > > > > > - Using **Pre-training Cost** as a measure of computational FLOPs, SpaceFormer (No. 12) and Uni-Mol (No. 6) have comparable costs, but SpaceFormer (No. 12) outperforms Uni-Mol (No. 6) across all 4 tasks.
> > > > > > > > > > > - Using the **number of empty points** as a measure of computational FLOPs, SpaceFormer with 200 points (No. 11) outperforms Uni-Mol (No. 6) on 3 out of 4 tasks.
> > > > > > > > > > >
> > > > > > > > > > > If the reviewer feels these results are insufficient, please let us know. We are willing to address further concerns if the additional cost and effort are reasonable.
> > > > > > > > > > >
> > > > > > > > > > > | No. | Model         | # Empty Points | R² ↓           | ZPVE ↓          | Cᵥ ↓           | HOMO ↓          | Pre-training Cost |
> > > > > > > > > > > |-----|---------------|----------------|----------------|-----------------|----------------|-----------------|--------------------|
> > > > > > > > > > > | 5   | Uni-Mol       | 100            | 3.3509         | 0.0006          | 0.1114         | 0.0020          | 17h                |
> > > > > > > > > > > | 6   | Uni-Mol       | 200            | 3.8193         | 0.0005          | 0.1145         | 0.0020          | 35h                |
> > > > > > > > > > > | 7   | Uni-Mol       | 400            | 4.3522         | 0.0004          | 0.1337         | 0.0023          | 96h                |
> > > > > > > > > > > | 10  | SpaceFormer   | 100*           | 3.3996         | 0.0004          | 0.0777         | 0.0024          | 17h                |
> > > > > > > > > > > | 11  | SpaceFormer   | 200*           | 3.2388         | 0.0004          | 0.0787         | 0.0024          | 19h                |
> > > > > > > > > > > | 12  | SpaceFormer   | 700*           | 2.8610         | 0.0005          | 0.0791         | 0.0018          | 32h                |
> > > > > > > > > > > | 13  | SpaceFormer   | 1300*          | 2.8431         | 0.0004          | 0.0725         | 0.0017          | 51h                |

---

> > > > > > > > > > > > ### Comment · Reviewer_jmPG · 2024-11-27
> > > > > > > > > > > >
> > > > > > > > > > > > Regarding the QM9 comparison: Sorry if this was not clear, but I did not suggest that you retrain SchNet or PaiNN on 40k samples. Neither SchNet or PaiNN are the current SOTA on the dataset (in my original reply, I merely pointed out that even PaiNN, an already three year old model, achieves lower errors than what was reported in the original QM9 results table in the paper). I understand the authors' argument that they cannot 1:1 compare to published numbers because they train only on 40k examples, but why do they not simply train their model under the established settings everybody else is using for benchmarks? This way, they could directly compare to published numbers. Retraining all other models under their settings is 1. much more work and 2. has huge potential for errors (the authors might not use optimal settings for the other models, or introduce errors in the training procedure). This is why I am saying the authors seem to "evade" proper comparisons. I am particularly concerned about the potential for errors, as this has evidently already happened: The first results reported on QM7-X in the authors' rebuttal had quite obvious problems, and to be honest, I am still skeptical about the fixed results that they now report, because models like SchNet and PaiNN achieve much lower errors in my experience on datasets like QM7-X, even when training on just ~10-20k data points and only on energies (I cannot point to the literature for exact numbers, because as mentioned above, there are usually established settings for publishing results on these benchmarks which the authors stray from).
> > > > > > > > > > > >
> > > > > > > > > > > > The comparison to Uni-Mol is similar: Neither the number of empty points used, nor the time for pre-training are a proper measure for establishing that models were compared under an iso-FLOP setting.  The authors claim that including empty space leads to better molecular representations, which seems rather extraordinary to me. As the saying goes, extraordinary claims require extraordinary evidence. It is therefore important to make sure that including empty space is actually what is leading to performance gains, and not merely an increase in effective FLOPs. I cannot understand why the authors do not simply run an experiment where an atom-based model like Uni-Mol is compared to SpaceFormer under iso-FLOP constraints.
> > > > > > > > > > > >
> > > > > > > > > > > > The gist is: The authors seem to consistently choose non-standard or contrived settings for comparisons to existing models, which make it hard to reach definite conclusions about their model. I already raised my score by 2 points in response to the changes made by the authors, as I value their time and acknowledge their effort. However, the other reviewers and I have made clear that the paper still has flaws, and I am unwilling to raise my score further unless these are properly addressed. As it stands, I follow reviewer rJDR's remark that I cannot recommend the publication of the paper in its present form, but I will also not object if the AC or other reviewers find the paper should be accepted anyway.

---

> ### Author Response · Authors · 2024-11-27
>
> We sincerely thank the reviewer for the additional comments.
>
> As stated in our initial response, we respectfully disagreed that a comparison with neural potential models (NPMs) is necessary to evaluate our contribution, as it falls beyond the scope of this paper. However, since the reviewer insisted, we conducted additional experiments to address this concern. Despite the significant effort involved in obtaining these results, the reviewer appears to doubt their validity. While we considered sharing the running logs to provide further evidence, we felt that it might not fully address the reviewer’s concerns. Nevertheless, we assure the reviewer that we have made every effort to address these points comprehensively and transparently.
>
> We understand that the reviewer may be highly familiar with NPM and has evaluated our work through that lens, expecting a comparison. However, in our opinion, such a comparison is not essential for evaluating the contributions of this work.
>
> ---
>
> ### Scope of Our Paper
>
> Our work focuses on **molecular representation learning (MRL)**, where large-scale data is used to pretrain a model, which is then evaluated on downstream tasks with very limited labeled data. In contrast, NPMs focus on predicting energy and force, primarily for molecular dynamics simulations.
>
> Moreover, our baselines—including Grover, GEM, Uni-Mol, and Mol-AE—do not include NPMs as baseline models. This demonstrates that NPMs are designed with fundamentally different goals compared to MRL models, and by default, our work does not aim to compare with NPMs. Extending MRL models to NPN tasks may be an interesting direction, but it is not the focus of our work.
>
> ---
>
> ### Our Designed Benchmark
>
> The evaluation of MRL models typically relies on limited labeled data due to the high cost of acquiring such data, especially for experimental properties. To reflect this, our benchmark includes subsampled QM9 to evaluate learning ability under limited labeled data conditions. While full QM9 is widely used in NPMs, the goal of our benchmark is very different from them.
>
> In our paper (Line 320), we elaborated on the rationale for proposing a new benchmark, noting the limitations of the widely used MoleculeNet dataset. Specifically:
> - Walters (2023) identified several issues with MoleculeNet.
> - Sun et al. (2022) demonstrated that MoleculeNet fails to adequately distinguish the performance of different molecular pretraining models.
>
> Our designed benchmark aims to address these shortcomings and provide a fair evaluation framework tailored to real-world MRL scenarios.
>
> ---
>
> ### Fair Comparison with Previous MRL Models
>
> To ensure an apple-to-apple comparison, we adopted the same pretraining data as the most recent works, Uni-Mol and Mol-AE. Additionally, we used the same model size as Uni-Mol and aligned all hyperparameters, including those for pretraining and fine-tuning, to ensure fairness in comparison.
>
> While reviewer rJDR raised concerns about our data split, we provided evidence demonstrating that our splitting method is highly similar to the one they recommended, further validating the robustness of our evaluation methodology.
>
> ---
>
> ### Summary
>
> We understand that the reviewer places significant emphasis on comparisons with NPMs, likely due to their expertise in this area. While we hold differing views on the necessity of such comparisons for this work, we value the reviewer’s input and have clarified our perspective to the best of our ability.
> Our goal in this reply is not to convince the reviewer to change their opinion but rather to clearly articulate our rationale and the scope of our work.
>
> We sincerely thank the reviewer again for their thoughtful effort and constructive feedback throughout the discussion.
>
> ---
>
> ### Additional Comments on iso-FLOP Comparison
>
> Regarding the reviewer’s statement, "Neither the number of empty points used, nor the time for pre-training are a proper measure for establishing that models were compared under an iso-FLOP setting," we remain unclear about what constitutes "a proper measure" as we thought that training cost or additional empty points is a direct measure.  We are not challenging the reviewer but are genuinely curious to understand their perspective and would greatly appreciate further clarification.
>
> updated:
>
> We now may understand the reviewer's point regarding the "iso-FLOP setting." There seems to be a potential misunderstanding that SpaceFormer’s model capacity is significantly larger than Uni-Mol’s. However, SpaceFormer and Uni-Mol share almost identical transformer architectures and model capacities, differing only in their positional encoding mechanisms. **As a result, when the number of input tokens is the same, the FLOPs required by SpaceFormer and Uni-Mol should be nearly identical**. Based on this, we considered the "number of empty points" a reasonable proxy for ensuring an "iso-FLOP setting.

---

### Official Review · Reviewer_rJDR · 2024-10-29

**Soundness:** 2
**Presentation:** 3
**Contribution:** 2
**Rating:** 5
**Confidence:** 4

**Summary:**

This paper presents SpaceFormer model for molecular representation learning which is based on transformer. Compared to previous methods which focus solely on atom/graph level information, SpaceFormer aims to capture spatial information more precisely by taking into account the 3D space beyond just atomic positions. For this purpose, the paper introduces methods for discretizing 3D space for computational efficiency while preserving precise atomic information and benchmarks the model on multiple downstream property prediction tasks.

**Strengths:**

1. The paper is well-motivated with physical principles and to my knowledge, introduces a novel MRL framework to explicitly capture the 3D spatial information beyond atomic positions. This allows the model to capture subtle spatial interactions between atoms that maybe crucial for molecular property.
2. To handle the infinite nature of 3D space and the computation costs associated with processing grid based representations of molecules, the authors propose extensions of several techniques known in different domains. For example,  the discretization procedure, the 3D angular positional encoding with RoPE and 3D radial positional encoding with RFF are sensible extensions of known techniques for the problem at hand.
3. The model is tested on multiple downstream tasks, demonstrating SpaceFormer’s effectiveness across a range of MRL applications.
4. The paper is well-written and easy to read.

**Weaknesses:**

1. Although grid discretization captures spatial information efficiently, it does inherently divide continuous 3D space into discrete cells. Could this lead to a loss of spatial continuity, especially when atoms or interactions lie near cell boundaries?
2. Can the proposed model handle irregular molecular shapes? The experiments considered in the paper are on simpler molecules. When extending it to irregularly shaped molecules, I wonder if the cuboid might include substantial regions without atoms, which could reduce efficiency?
3. The approach relies heavily on accurate atomic positions and bond distances, which might not always be readily available or precise for all molecular structures, especially in experimental or noisy datasets. Have the authors studied this aspect?
4. Do I understand it correctly that the proposed method uses a fixed grid size within the effective cuboid, which may not adapt well to regions with varying atomic densities. For densely packed regions, the chosen grid size might be too coarse, while for sparsely packed regions, it may be overly fine?
5. The paper studies the model performance under two kinds of data splits: In-Distribution Split, where the sets are randomly partitioned, and Out-of-Distribution Split, where the sets are divided based on fingerprint similarity. The Out-of-Distribution Split is somewhat realistic setting however in real-world drug discovery applications, much more common splitting strategies are scaffold split or temporal split. I would be curious to see how the model performs under such settings.
6. The paper mentions several related works but the baselines compared against SpaceFormer are rather limited. How does the proposed model compare against works such as NoisyNodes, GraphMVP, 3D-InfoMax etc.?

**Questions:**

I have listed my questions in the weaknesses section already. To summarize:
1. Could the discretization process lead to a loss of spatial continuity, particularly when atoms or interactions lie near cell boundaries? How does the model account for this, if at all?
2. Can the proposed model efficiently handle irregularly shaped or large molecules?
3. Given that the model relies heavily on accurate atomic positions and bond distances, how does it perform on experimental or noisy datasets where such information may be less precise or unavailable?
4. Does the use of a fixed grid size within the effective cuboid limit the model’s adaptability to regions of varying atomic density? I may have misunderstood this aspect.
5. Could the authors comment on how SpaceFormer might perform under more real-world splitting strategies (temporal and scaffold splits) for training and evaluation?
6. Could the authors provide comparisons to how the model performs relative to other recent works, such as NoisyNodes, GraphMVP, or 3D-InfoMax?

---

> ### Author Response · Authors · 2024-11-18
> **response to Reviewer rJDR**
>
> We sincerely thank the reviewer for the insightful comments. Below, we provide detailed answers to your questions:
>
> 1. **On spatial continuity:**
>
> We appreciate the concern regarding the potential disruption of spatial continuity caused by gridding, especially for atoms near cell boundaries. As described in Section 3.1, we address this issue by preserving the *precise* atomic positions, regardless of which cells the atoms belong to. These precise, continuous atomic positions are used consistently throughout the model, ensuring that gridding does not affect atom spatial information.
>
> Additionally, as an implementation detail, we introduce a small translation noise to all atomic positions before gridding. This softens the boundaries between grid cells and mitigates potential issues with spatial continuity, providing smoother transitions and improving overall robustness.
>
> 2. **On handling  irregularly shaped or large molecules**:
>
> Our grid sampling strategy (Section 3.2) is designed to handle this issue. Specifically, the strategy samples fewer cells in coarse regions and more cells in fine regions with higher atomic density. For regions without atoms, only a small number of cells are sampled, ensuring computational efficiency while capturing the relevant spatial structure.
>
> 3. **On Reliance on Accurate Atomic Positions**:
>
> Both the pretraining and downstream tasks do not rely on highly accurate atomic positions. We adopt the approach from Uni-Mol, using RDKit -- a widely used chemical toolkit -- to generate atomic positions when accurate atomic positions are not provided. RDKit efficiently generates positions in just a few milliseconds per molecule, making this a practical solution. Therefore, the lack of high-precision atomic positions does not limit the applicability of our method.
>
> 4. **On the "fixed grid size"**:
>
> If we correctly interpret the reviewer’s concern, it pertains to the fixed grid cell edge length. While we use a fixed edge length in our design, we initially experimented with an adaptive gridding strategy based on an octree structure. This approach merged adjacent grid cells hierarchically, creating fewer, larger cells in coarse regions and more, smaller cells in fine regions with higher atomic density.
> Conceptually, this is similar to our importance sampling strategy, which also allocates fewer cells to coarse regions and more to fine regions. However, our sampling strategy offers better control over the total number of cells, providing greater flexibility and efficiency. For this reason, we adopted it in the final design.
>
> 5. **On the data split method**:
>
> Thank you for your insightful feedback regarding splitting strategies. We chose the Out-of-Distribution (OOD) split based on fingerprint similarity to investigate the model’s ability to generalize to chemically distinct compounds. Fingerprints, which encode molecular structures into bit vectors, are widely used in cheminformatics for tasks like compound similarity and diversity evaluation. As such, OOD splits based on fingerprint similarity offer a meaningful way to assess how well a model performs when encountering structurally dissimilar compounds.
> We agree that scaffold and temporal splits are highly relevant and frequently employed in drug discovery. While fingerprint-based splits provide an initial evaluation of robustness, scaffold splits capture structural novelty, and temporal splits simulate real-world scenarios where newer compounds must be predicted based on historical data. In the revised manuscript, we will expand the discussion to address these complementary approaches and propose their inclusion as future work.
>
> 6. **On more baseline models**:
>
> Thank you for your thoughtful comment regarding the comparison of SpaceFormer with other recent works. In this study, we have already compared SpaceFormer with both graph-based and 3D representation molecular pretraining methods, ensuring that the baselines are representative of the state of the art in these paradigms.
> we acknowledge the value of extending comparisons to include recent advances like NoisyNodes, GraphMVP, and 3D-InfoMax. While additional experiments were not feasible due to time and resource constraints, we will expand the discussion and include further experiments in the revised manuscript to provide a broader evaluation.

---

> > ### Comment · Reviewer_rJDR · 2024-11-26
> > **Thank you for your response**
> >
> > I thank the authors for their response. However, I feel that my major concerns regarding experiments on data splitting (temporal and scaffold splits) remain unaddressed. The random OOD split is not representative for real-world drug discovery applications and in my personal experience does not work well in real-world scenarios. In addition, the authors also did not present any comparions with recent baselines such as NoisyNodes, 3D-InforMax etc. which do consider 3D structural information and hence are relevant to understand the contributions of the proposed work. Just adding discussions about these methods without justifying why these methods have not been compared against is not sufficient in my opinion given that there were 2 full weeks for adding comparisons.
> > Since my major concerns remain unaddressed, unfortunately I cannot change my score and won't be able to recommend this work for acceptance at this time.

---

> > > ### Author Response · Authors · 2024-11-26
> > >
> > > The main contribution of this paper is to demonstrate that incorporating empty space can further enhance molecular representation learning models. To substantiate this, we use the same pretraining dataset as prior works (Uni-Mol and Mol-AE) and adopt identical downstream task settings for comparison. This ensures an apple-to-apple comparison, and we are confident that the current experimental results fully support our contribution.
> > >
> > > The baselines you mentioned (NoisyNodes, 3D-InforMax) use a different pretraining dataset, including them would not provide an apple-to-apple comparison and would not meaningfully validate our proposed approach.
> > >
> > > Additionally, we believe that two weeks is insufficient to implement these high-burden additional experiments. Preparing the rebuttal, revising the paper, and conducting other requested experiments have already occupied our capacity during this limited timeframe. We respectfully ask the reviewer to evaluate our work reasonably based on its core contributions, rather than requiring infeasible or less meaningful additional experiments.

---

> ### Comment · Reviewer_rJDR · 2024-11-26
> **Thank you!**
>
> Thank you for your response. I do not mean to come across as asking for infeasible or less meaningful experiments and I am not sure why the authors have this impression. The authors could have simply run pre-training for one property or showed partial experiments with different data splitting strategies if time was an issue. Especially when I mentioned that in real-world drug discovery random splitting is not what practitioners do and that is a big issue with building ML models.
>
> Regarding baselines, I understand that NoisyNodes, 3D-InfoMax etc. are different pre-training strategies. My question is if given a choice in real-world setting, why and when should practitioners use the presented method vs the other pre-training methods, especially when no comparison or discussion is presented about merits and demerits of each? I assume that the authors will agree that the goal of any such work would be to help the practitioners and help drug discovery applications. Hence, for a strategy to be useful in practice, we should know when to use which method and without a comparison with methods that are known to work well, it is hard to do so.
>
> I understand that preparing rebuttals, revising the paper and conducting experiments takes time and I fully understand the authors. Anyways, I will respectfully suggest that in future authors may choose to do these experiments. I understand that reviewer jmPG who have been highly engaged in discussion and raised good points also agree with my concerns. Anyways, I am happy to be overruled and my suggestions and ratings discounted by AC and other reviewers if they consider them infeasible or less meaningful too. Good luck!

---

> > ### Author Response · Authors · 2024-11-26
> >
> > While the reviewer insists that a scaffold split is necessary, we respectfully hold a different opinion, which we explained in our response 8 days ago. Unfortunately, we did not receive further feedback on this point during that time, leaving us without clarification on the reviewer’s stance.
> >
> > Even now, we believe that a scaffold split is not essential to demonstrate our contribution. However, if the reviewer had emphasized this earlier, we would have had sufficient time to conduct the additional experiments. For example, Reviewer jmPG promptly replied to our comments, allowing us to run additional experiments to address their concerns efficiently.
> >
> > Regarding the baselines, we would like to clarify further. Our comment referred specifically to the pretraining dataset, not the pretraining strategy. NoisyNodes uses the PCQM dataset, while 3D-Informax uses QM9 and GEOM-Drug, both of which differ from the dataset used for our baseline.
> >
> > As for pretraining strategies, Uni-Mol employs a method similar to NoisyNodes by leveraging position denoising. Meanwhile, 3D-Informax is fundamentally a 2D graph model augmented with 3D data, making it less directly relevant to our work.

---

> ### Comment · Reviewer_rJDR · 2024-11-27
>
> Thank you for pointing out that I did not respond to your point about scaffold split 8 days ago. Please note that in my initial review, I unambiguously requested for "in real-world drug discovery applications, much more common splitting strategies are scaffold split or temporal split. I would be curious to see how the model performs under such settings." In their response, the authors simply acknowledged that these splits are sensible but no experiments were shown. However, if the authors believe I was not clear enough in my initial review, I apologize. Also if the authors believe that scaffold or temporal splits are not necessary for their work, I simply disagree. I gave my reasons and it's ok that the authors disagree with my reasoning.
>
> As I said, I cannot champion the acceptance of this work in its current form even though I think it proposes an interesting idea, but I will not oppose its acceptance if AC and other reviewers wish to and I am happy for my reasons, reviews and ratings to be discounted and discarded from consideration. Good luck!

---

> > ### Author Response · Authors · 2024-11-27
> >
> > It appears that we have addressed the reviewer’s concern regarding the addition of more baselines.
> >
> > Here, we provide additional evidence about the effectiveness of our OOD split.
> >
> > Specifically, we calculated the scaffold overlapping rates of the validation and test sets relative to the training set in our OOD split, as shown in the following table. From the table, we observe that for most datasets (except QM9), the scaffold overlapping rate in the validation and test sets is only 0.4–1.6%. This is because the fingerprint method we use encodes structural information, including scaffold features.
> >
> > In summary, although our OOD splits are derived using a different approach, they are very similar to scaffold splits in practice. Therefore, we believe our current results are valid for scenarios commonly encountered in real-world drug discovery applications.
> >
> > QM9 is an exception, with a scaffold overlapping rate of approximately 10%. To address this, we are currently re-running the experiment on QM9’s HOMO prediction task with scaffold splits. We will update the results here as soon as the experiments are completed.
> >
> >
> > | Dataset                | Task Type      |#Samples | #Train Scaffold | #Valid Scaffold in Train | Valid / Train | #Test Scaffold in Train | Test / Train|
> > |------------------------|----------------|------|------|----------------|--------|----------------|-------- |
> > | **qm9**          | Regression     |  40,000    |5,967 | 517            | 0.086 | 700            | 0.117 |
> > | **bbbp**               | Classification |   1,965   |925  | 11             | 0.011  | 10             | 0.010  |
> > | **bace**               | Classification |  1,513    |589  | 3              | 0.005  | 4              | 0.006  |
> > | **HLM**     | Regression     | 3,087 |1,866 | 16             | 0.007  | 32             | 0.015  |
> > | **MME**  | Regression     |  2,642 | 2,173 | 7             | 0.004  | 28             | 0.015  |
> > | **SOLU**  | Regression     | 2,713  | 1,529 | 24             | 0.016  | 23             | 0.015  |

---

> > > ### Author Response · Authors · 2024-11-29
> > >
> > > We sincerely thank the reviewer for their patience. We have completed the results for QM9's HOMO prediction using the scaffold split, as shown in the following table. The results clearly demonstrate that our model outperforms the previous baselines even under the scaffold split. Furthermore, across different split methods -- random, OOD, or scaffold -- the performance ranking remains consistent.
> > >
> > > In summary, we believe these results sufficiently address the reviewer’s concerns regarding data splitting. Specifically, the data splits in 5 out of 6 datasets are highly similar to the scaffold split, with only 0.4–1.6% scaffold overlapping. For the remaining dataset (QM9), we conducted an additional experiment using the scaffold split, and the results confirm that the performance ranking is consistent with our original split method.
> > >
> > >
> > > | Model | Our Random Split | Our OOD Split | Scaffold Split |
> > > | -------| ----------|----------------| -------------------|
> > > Uni-Mol | 0.0019 | 0.0020 | 0.0028 |
> > > Mol-AE | 0.0020 | 0.0023 | 0.0033 |
> > > SpaceFormer | **0.0016**  | **0.0017** |  **0.0021** |

---

### Official Review · Reviewer_ApRY · 2024-10-30

**Soundness:** 2
**Presentation:** 3
**Contribution:** 2
**Rating:** 5
**Confidence:** 4

**Summary:**

I appreciate the opportunity to review this manuscript on "More Space Is All You Need: Revisiting Molecular Representation Learning."
The paper presents SpaceFormer, a transformer-based framework for molecular representation learning that aims to demonstrate how leveraging 3D space beyond atoms can enhance molecular representations. The work introduces grid-based sampling and combines RoPE and RFF techniques.

**Strengths:**

- Novel combination of RoPE and RFF techniques
- Innovative grid-based sampling approach for spatial representation
- Interesting comparison with Uni-Mol using randomly sampled empty points
- Technical innovation in spatial representation methodology

**Weaknesses:**

Reproducibility Concerns
A critical issue that must be addressed immediately is the lack of code availability. This significantly hampers reproducibility - a cornerstone of scientific research. The code should be considered an extension of the experimental setup, and its absence makes proper evaluation challenging. Without access to the implementation details and the ability to reproduce the results, it becomes difficult to fully validate the claims and insights presented in the paper.

Fundamental Research Question and Historical Context
A significant concern lies in the paper's central research question: "Will leveraging the 3D space beyond atoms improve molecular representation learning?" This framing overlooks substantial existing work in the field. Models like ANI and SchNet have already definitively demonstrated the value of considering complete 3D space through their use of Behler-Parrinello symmetry functions and sophisticated spatial representations. These established approaches already:
- Capture the complete local chemical environment
- Incorporate radial and angular features describing inter-atomic spaces
- Map the entire local 3D space around atoms into feature vectors
- Consider continuous space rather than just discrete atomic positions
Therefore, the more appropriate research question should focus on finding novel, potentially more efficient ways to represent and process 3D spatial information, rather than questioning whether such spatial consideration is beneficial.

Technical Innovation and Comparative Analysis
While the paper's technical approach using grid-based sampling shows innovation, the comparative analysis raises concerns. The baseline model selection appears inadequate, particularly given the availability of established models that already leverage 3D space effectively. The manuscript's results show that SchNet frequently outperforms the proposed SpaceFormer model on the QM9 dataset. The absence of comparisons with models like ANI2 or SchNet requires justification, especially given their relevant capabilities and established performance.

Technical Implementation Concerns
The use of PCA for coordinate system alignment presents potential issues that need addressing. PCA axes can flip when eigenvalues are similar, potentially leading to arbitrary axis assignments. This could be particularly problematic for molecules with different conformations. The manuscript should address:
- Whether this possibility was investigated
- The proportion of affected molecules in the dataset
- Potential solutions, such as augmentation approaches involving random rotation and translation of voxel boxes

Visualization and Explainability
Two significant improvements would enhance the manuscript:
- The latent space structure after pre-training needs thorough examination through ablation studies. Understanding what the model actually learns would improve transparency and help readers better grasp the embeddings' potential applications in downstream tasks.
- Figure 1's representation of the gridding and grid sampling approach should more clearly demonstrate its 3D nature to avoid ambiguity.

**Questions:**

1. How do you address PCA axis flips when eigenvalues are similar?
2. Why were comparisons with ANI2 and other established models omitted?
3. Can you provide ablation studies examining the latent space structure (what has been learned)?
4. How will you ensure reproducibility through code availability?
5. What percentage of molecules in your dataset are affected by PCA alignment issues?
6. How do you justify the model's performance given e.g. SchNet's superior results on QM9?

(For more details look at the strengths and weaknesses above)

---

> ### Author Response · Authors · 2024-11-18
> **response to Reviewer ApRY**
>
> We sincerely thank the reviewer for the insightful comments. Below, we provide detailed responses to your questions:
>
> 1. **On the Claim That “ANI and SchNet Have Already Definitively Demonstrated the Value of Considering Complete 3D Space”:**
>
> We respectfully disagree with this claim. Both ANI and SchNet are atom-based models that take only atoms as input. Additionally, the “Behler-Parrinello symmetry” feature vector used in these methods considers only local neighborhood atoms around each atom, rather than the entire 3D space. While methods like SchNet employ continuous kernels, such as the RBF kernel, these primarily encode pairwise atom distances and only apply to atom-pair interactions. As such, these models may implicitly capture limited spatial information beyond atoms but do not explicitly consider or demonstrate the value of the complete 3D space, particularly the regions beyond atoms.
>
> In contrast, SpaceFormer explicitly incorporates non-atom cells into its input, allowing it to directly utilize information from the full 3D space. Our experimental results show that this approach consistently outperforms atom-only models, emphasizing the benefits of explicitly modeling non-atomic regions.
>
> 2. **On the "absence of comparisons with models like ANI2 or SchNet":**
>
> We appreciate the reviewer’s suggestion to include comparisons with SchNet and ANI2, as these models indeed demonstrate strong performance in their respective domains. However, we believe their focus and application differ significantly from the scope of our work.
>
> SchNet and ANI2 are *neural potential models* designed specifically for energy and force prediction tasks, such as molecular dynamics simulations. Their primary objective is to approximate potential energy surfaces with high accuracy for quantum chemistry and molecular mechanics applications.
> In contrast, this paper focuses on *molecular representation learning* (MRL), where the goal is to pretrain on large-scale unlabeled molecular data and transfer the learned representations to downstream tasks, such as property prediction, with limited labeled data. While there are conceptual overlaps, MRL models and neural potential models are designed for fundamentally different applications and are evaluated using distinct protocols.
>
> Furthermore, most MRL studies, including the baselines in our manuscript, do not consider neural potential models as direct baselines because their training paradigms and evaluation metrics are not aligned with MRL objectives. Consequently, comparisons with SchNet and ANI2 would not provide meaningful insights into the effectiveness of our approach for MRL tasks.
>
>
> 3. **On the claim that "SchNet frequently outperforms the proposed SpaceFormer model on the QM9 dataset"**:
>
> It is important to note that the QM9 dataset used in this paper is a **subsampled** version containing **40k samples**, whereas the original QM9 dataset consists of **133k samples**. Therefore, direct comparison of our results with those reported in previous papers using the full dataset would be unfair. The subsampling in our study is intentional, as the paper focuses on evaluating the performance of a pretrained MRL model on downstream tasks with limited data. In real-world applications, the availability of labeled data is often constrained due to the high costs of computation and wet-lab experiments. This setup, therefore, enables a more realistic assessment of the model’s effectiveness in practical scenarios.
>
> 4.  **On the instability of PCA:**
>
> The primary instability of PCA arises from the arbitrary orientation of axes and the potential for axis swapping. To address these issues, we incorporate atom weights to determine axis directions. Specifically, atoms are projected onto each PCA axis, and the mean atom weight on either side is calculated. The side with the larger average atom weight is designated as the positive direction, reducing ambiguity and ensuring consistency.
> Additionally, as a safeguard, we verify the validity of the generated coordinate system by checking for orthogonality and right-handedness. If the system fails these checks, we fall back to the original coordinate system to maintain robustness.
>
> It is also important to note that we use PCA solely to define an effective cuboid for grid size reduction. As such, axis flips have minimal impact on the grid structure and, consequently, only a negligible effect on our method. In fact, during early experiments, we applied PCA without the above stability optimizations and observed no difference in performance after addressing these issues.
>
> 5. **Reproducibility**:
>
> Very sorry for missing Reproducibility Statement as it is not in ICLR template. We will open source all our code after acceptance.

---

> ### Author Response · Authors · 2024-11-23
> **Additional comparison with neural potential models**
>
> **Update to the Reply 2:**
>
> We have conducted additional comparisons with neural potential models, such as SchNet and PaiNN, for both property prediction and energy/force prediction. The results can be found in our replies: https://openreview.net/forum?id=LBsr2llHz0&noteId=AL0f3opV3m and https://openreview.net/forum?id=LBsr2llHz0&noteId=Yv5WQCVQ7H.
>
> These experiments required substantial time and effort, demonstrating our commitment to thoroughly addressing the reviewer’s concerns. We sincerely hope the reviewer will recognize our dedication and reconsider their evaluation.

---

### Official Review · Reviewer_kPDK · 2024-11-03

**Soundness:** 2
**Presentation:** 2
**Contribution:** 2
**Rating:** 5
**Confidence:** 4

**Summary:**

The paper introduces a novel architecture for point-cloud molecular data based on the PCA normalization followed by a space transformer. The results seems to be on the SOTA level. However, the Reproducibility Statement is missing and the code is not attached. It's therefore very hard to evaluate the model.
Technically, the innovations are controversial. A lot of stress in the paper is put on SE3 invariance with respect to global rotation/translation. However, after the PCA, the input representation is already invariant. Another stress is put into absolute positional encoding in SE3. Here, no prior work on geometric attention/transformers is cited, where the same problem has been already solved for point cloud and graph representations. Also, the proposed solution is not rigourosly correct. Finally, I question the gridding technique, why it is needed here as the main blocks operate on continuous positions? Again, sampling additional points in geometric/graph attention and transformer architectures have been well studied in the literature, and must be properly cited.

**Strengths:**

The paper is well written and presents a model that outperforms the state of the art. The author even retrained and experimented with the SOTA architecture Uni-Mol.

**Weaknesses:**

The Effective Cuboid for Gridding -- the PCA vectors have arbitrary sign, and also uncertainty in the directions if two or more eigenvalues are close in value. As a result, you may have instability, at least with respect to the sign of the frame. This issue has to be at least discussed. Ideally, more experiments need to be run with sign flip of the PCA vectors.

3D Angular Positional Encoding with RoPE is very misleading. You seem to operate on directions from an arbitrarily chosen coordinate center, and measure relative angles with respect to this center. Does it make any sense? It wouldn't make sense even in 2D... Could you please draw the 2D geometry and explain what will be this angle?
The relative angle is indeed invariant to an in-plane rotation. However, it will change with a different choice of rotation plane, and it will also change with a different choice of the coordinate system (the center of rotation).

The Gaussian kernel approximation has to be validated and explained better, it's very nontrivial why a non-periodic function can be decomposed through products of periodic functions. Even the original paper states "For the Gaussian kernel, k(δ) is not convex, so k is not everywhere positive and δk(δ) is not a probability distribution, so this procedure does not yield a random map for the Gaussian."

More generally, your representation is rather similar to the point-cloud representation with some additional sampling points (which have been studied in the literature), as you are using continuous coordinates as input and full-length attention. Could you please clearly identify the benefits of additionally encoding grid-related parameters?

Sampling empty grids is similar to constructing virtual or aggregation nodes in point-cloud or graph representations. It will be useful to discuss it. Another analogy is stochastic sampling strategy to compute attention better than in N^2, which is used in some recent architectures, please discuss it too.

**Questions:**

We incorporate FlashAttention (Dao et al., 2022), reducing the memory cost from O(n2) to linear complexity - this statement is not fully correct. The FalshAttention paper states : We analyze the IO complexity [1] of FlashAttention, proving that it requires 𝑂(𝑁^2𝑑^2 𝑀^-1) HBM accesses where 𝑑 is the head dimension and 𝑀 is the size of SRAM.
So, it will only scale linearly when the cache size is bigger than Nd (or, formally, is a function of (ND). Which is not the case if the size of your system grows bigger.
The linear scaling is questionable even looking at the experiments (table 6). Models no 13-14 have an increase in the number of cells by 15%, but the time doubles -- this probably indicates the regime when you saturate the SRAM. Please correct the statements.

"Second, in 3D MRL models, SE(3)- invariant positional encoding" - you already claim to work in invariant frames (after the PCA), the SE3 invariance is not needed here it seems. Motivate the SE3 invariance better please. Maybe you simply gain in performance because of the PCA normalization?

"While the above RoPE-based encoding ... it may be unstable when the entire system undergoes global rotations.", well, it is not supposed to be unstable by design, this was the main idea of RoPE. If it is unstable, please clearly explain why. More generally, you are already using the pose normalization method (PCA), you are INVARIANT to global rotations and translations. So, I believe, this argument is not valid.

"utilizing a 3D positional encoding that is invariant to global rotation and translation" - again, you are in the PCA frames, why do you discuss it?.

In 3D, the actual rotation depends on the sequence of 3 individual rotations (see Euler angle conventions, for example).. Which one do you use? How do you decompose a 3D rotation into 3 rotations about different axes? Does it make any difference? How stable the result will be if you remove the PCA normalization?

Some minor points:
"which is inspired from microscopic physics where regions near atoms are more crucial." - more crucial for what?
"regions close to atoms exhibit higher electron density" - better say close to atom centers, or to nuclei centers
"in computational simulations, coarse-graining is commonly applied to regions farther from atoms to reduce computational cost." - please explain it better and more rigorously
"Therefore, our sampling strategy is based on the distance from the nearest atom cells." - you can motivate it in a simpler way.

Can the learned representations (Figure 2) be connected to the way you sample empty space using the importance sampling? What will be the representations using uniform or other types of samplings?

---

> ### Author Response · Authors · 2024-11-18
> **response to Reviewer kPDK (1/2)**
>
> We sincerely thank the reviewer for their detailed comments. However, we believe there are some misunderstandings regarding our method, particularly concerning PCA and the proposed positional encoding. We address these points below and will revise the paper to make these aspects clearer.
>
> 1.  **On PCA and SE(3) Invariance:**
>
> PCA is not intended to handle SE(3) invariance in our method. Its motivation is solely to reduce the grid size, *not to* (and *cannot*) address SE(3) invariance. In molecular data, the inherent symmetry often causes PCA to fail to produce a unique coordinate system, thereby making it unreliable for ensuring SE(3) invariance. Furthermore, if PCA could perfectly handle SE(3) invariance, there would *not* be hundreds of papers addressing SE(3)-invariance/equivariance challenges. The reviewer’s comemnt “PCA vectors have arbitrary signs and uncertainties in direction when eigenvalues are close” further supports this.
> Consequently, since PCA cannot ensure SE(3) invariance, our proposed positional encoding, which captures pairwise distances using Random Fourier Features, is essential for providing SE(3)-invariant input features.
>
> 2. **On Random Fourier Features and Gaussian Kernels:**
>
> Approximating Gaussian kernels with Random Fourier Features (RFF) is a well-established and validated approach. Its original paper [1], which received the Test-of-Time Award at NeurIPS 2017, demonstrates the effectiveness of this technique. The reviewer's comment that “For the Gaussian kernel, k(δ) is not convex, so k is not everywhere positive and δk(δ) is not a probability distribution…” refers to the Random **Binning** Features approximation, another approach in that paper, not the Random **Fourier** Features  method we use. For further details on Gaussian kernels approximated with RFF, we kindly direct you to Section 3 of paper [1].
>
> 3. **On 3D RoPE and Pairwise Relative Position Encoding:**
>
> We apologize if the term “3D Angular Positional Encoding with RoPE” caused any confusion. The term “Angular” might indeed be misleading; “Directional” is a more accurate description in this context. Below, we provide a clearer explanation, starting from the concept of RoPE.
> Rotary Positional Embedding (RoPE) is a widely used method for encoding pairwise relative positions. Originally proposed for 1D sequences in language models, RoPE has since been successfully adapted for 2D image tasks [2].
> In our work, 3D RoPE extends this concept to encode the pairwise relative position of two points in 3D space.  For points A and B, with  coordinates (x1, y1, z1)and (x2, y2, z2), their relative position  is represented as $\vec{AB} = B - A = (x2-x1, y2-y1, z2-z1)$. Since vector $\vec{AB}$ contains directional information, we refer to it as “Directional Positional Encoding.” This direction is inherently dependent on the coordinate system and varies with rotation, which is precisely our goal. By doing so, the model can capture coordinate system information and better predict point positions within it. This approach is consistent with existing SE(3)-equivariance models, such as EGNN [3], which also leverage relative positional information for prediction.
>
> To summary, we believe that our proposed method is rigorously correct, and our ablation study on positional encoding (Sec. 4.3) also demonstrated the effectiveness of the proposed method. We hope these clarifications resolve the misunderstandings and please let us know if you have any other questions.
>
> Reference:
> [1] Rahimi, Ali, and Benjamin Recht. "Random features for large-scale kernel machines." Advances in neural information processing systems 20 (2007).
>
> [2] Heo, Byeongho, Song Park, Dongyoon Han, and Sangdoo Yun. "Rotary Position Embedding for Vision Transformer." arXiv e-prints (2024): arXiv-2403.
>
> [3] Satorras, Vıctor Garcia, Emiel Hoogeboom, and Max Welling. "E (n) equivariant graph neural networks." In International conference on machine learning, pp. 9323-9332. PMLR, 2021.

---

> > ### Comment · Reviewer_kPDK · 2024-11-24
> > **Further comment on RoPE**
> >
> > I thank the authors for their responses.
> >
> > Regarding the 2D angular RoPE - my question is still valid. Your directions are fixed by the PCA normalization, that's why you have a certain stability. You may try removing the PCA step and present the results with and without RoPE. This will be a proper demonstration of the proposed positional encoding.
> >
> > Secondly, the angular difference of directions after RoPe does not correspond to the direction of the xi-xj vector. You can see it by, for example, varying the norm of the xj point. The relative angular distance between xi and xj will keep constant, but the angular position of xi-xj will change.
> >
> > Finally, without the PCA step, you (the network) would not be able to guess the 3D orientation from the combination of 2D rotations (rotations do not commute).
> >
> > As a result, I believe, the PCA step is important for the performance, as there is no theoretical foundation of the useful information about xi-xj orientation otherwise. Please remove the PCA step and do a proper ablation study (with rotated samples; with and without the RoPE).

---

> ### Author Response · Authors · 2024-11-18
> **response to Reviewer kPDK (2/2)**
>
> Belows are the responses to your other comments.
>
> 4. **On the necessity of gridding:**
>
> Gridding becomes essential when considering both atoms and empty space, as SpaceFormer does. At a high level, gridding in SpaceFormer serves as an efficient strategy to sample empty points by constraining them to the centers of grid cells. The ablation study in Section 4.4 of the paper explicitly confirms the necessity of gridding. Without gridding, previous atom-based models sample empty points from a large, continuous 3D space, which fails to improve performance even with an increased number of sampled points. In contrast, SpaceFormer leverages gridding to define empty cells systematically, enabling it to utilize information from empty space more effectively.
>
> 5. **On the comparison with additional virtul point used in point-could domain:**
>
> We acknowledge that we overlooked citing and discussing related works on point clouds, as our primary focus was on molecular representation learning (MRL). We thank the reviewer for pointing this out and have since conducted further investigation. In the point cloud domain, virtual or grid points are often introduced as intermediate representations to address challenges such as resolution inconsistencies (e.g., between sensors or distant vs. nearby objects), enhance feature aggregation, or increase sampling density. These intermediate points act as bridges to improve geometric consistency and feature representation.
>
> In contrast, MRL tasks primarily focus on predicting molecular properties, where there is no natural need for intermediate representations like virtual points. Moreover, as demonstrated in Section 4.4, simply adding virtual points fails to enhance the performance of existing models. This distinction explains why prior MRL methods have not adopted virtual points.
>
> Our contribution lies in recognizing this gap and proposing a framework that effectively leverages empty space information, addressing an overlooked aspect of MRL. The validity and utility of our approach are supported by our experiments. We will revise the manuscript to include a discussion of related works on point clouds to clarify this distinction.
>
> 6. **On the instability of PCA:**
>
> The primary instability of PCA arises from the arbitrary orientation of axes and the potential for axis swapping. To address these issues, we incorporate atom weights to determine axis directions. Specifically, atoms are projected onto each PCA axis, and the mean atom weight on either side is calculated. The side with the larger average atom weight is designated as the positive direction, reducing ambiguity and ensuring consistency.
> Additionally, as a safeguard, we verify the validity of the generated coordinate system by checking for orthogonality and right-handedness. If the system fails these checks, we fall back to the original coordinate system to maintain robustness.
>
> It is also important to note that we use PCA solely to define an effective cuboid for grid size reduction. As such, axis flips have minimal impact on the grid structure and, consequently, only a negligible effect on our method. In fact, during early experiments, we applied PCA without the above stability optimizations and observed no difference in performance after addressing these issues.
>
> 7. **On the Sampling Strategy of Learned Representations in Figure 2:**
>
> As stated in the paper (Line 520 in the first submitted version), the learned representation shown corresponds to the Full-Grid SpaceFormer encoder (No. 2 in Table 4). No sampling strategy is applied to ensure a fair comparison with the electron density.
>
> 8. **Reproducibility**
>
> Very sorry for missing Reproducibility Statement as it is not in ICLR template. We will open source all our code after acceptance.

---

> ### Author Response · Authors · 2024-11-24
>
> Thank you for your response and valuable feedback. We hope our previous responses have addressed your other concerns.
>
> Regarding your request for an additional ablation study, due to the limited time before the discussion ends, we are afraid we may not be able to complete them in time. However, we believe our current results are sufficient to demonstrate the differences between PCA and RoPE in terms of their contributions to the final performance. Specifically, by combining the results from Table 3 and Table 5, we present the following observations:
>
> | PCA | RoPE | RFF | R2 ↓ | ZPVE | ↓ Cv ↓ | HOMO ↓ |
> | --- | ---- | ----| ----| ---- | ----| ----|
> | yes | yes |  yes |2.8363 | 0.0003 | 0.0675 | 0.0017 |
> | no | yes |  yes |3.3088 | 0.0004 | 0.0708 | 0.0018 |
> | yes | no |  no  | 3.7104 | 0.0004 | 0.1407 | 0.0022 |
>
> From these results, it is clear that PCA does not significantly contribute to the final performance, while the proposed positional encoding techniques (RoPE and RFF) play a much larger role in improving the final performance .
>
>
> Regarding your second point: If we only encode $x_i - x_j$, the direction $\frac{x_i - x_j}{\||x_i - x_j\||}$ is indeed not explicitly represented. However, in our approach, we also encode the distance $\||x_i - x_j\||$ using RFF. This allows the model to implicitly capture the direction $\frac{x_i - x_j}{\||x_i - x_j\||}$ through the combined encoding of $x_i - x_j$ (by 3D RoPE) and $\||x_i - x_j\||$ (by RFF).
>
> We hope our response above addresses your concern regarding PCA. Please let us know if you have any additional questions or require further clarification.

---

> > ### Comment · Reviewer_kPDK · 2024-11-24
> > **Further on RoPE**
> >
> > You cannot capture the direction  $\frac{x_i - x_j}{||x_i - x_j||}$ by only an angle $\alpha(x_i,x_j)$ and the norm $||x_i - x_j||$. This information is not sufficient. By analogy, you cannot define a plane triangle by only knowing its side and the opposite angle - there are infinitely many solutions.
> >
> > What happens in practice I believe - your are memorizing geometries in the PCA frames. The table shows it very clearly, without PCA results drop down. And probably you could have used another type of positional encoding here, RoPE does not provide any analytical guarantees for your case.

---

> > > ### Author Response · Authors · 2024-11-25
> > >
> > > It seems we have identified the misunderstanding. It seems the reviewer assumed that our 3D RoPE is used to encode the angle ∠OAB, where O is the origin, and A and B are two points in 3D space. However, 3D RoPE directly encodes the relative position vector $x_i - x_j$, rather than the angle.

---

> > > > ### Author Response · Authors · 2024-11-25
> > > >
> > > > We hope the above clarification addresses the reviewer's concerns regarding RoPE.
> > > >
> > > > **Regarding the comment: “..., you are memorizing geometries in the PCA frames. The table shows it very clearly, without PCA results drop down. And probably you could have used another type of positional encoding here, ...”**:
> > > >
> > > > It is true that performance decreases without PCA; however, another factor must be considered: without PCA, the grid size becomes larger, which may have an impact on performance.
> > > >
> > > > Furthermore, if the model were truly "memorizing geometries in the PCA frames," the "PCA only" setting would exhibit strong performance as well. In the "PCA only" configuration, positional information is incorporated by adding the linear projection of the 3D coordinates to the input embeddings. Despite this, the performance does not show significant improvement, indicating that the model's effectiveness is not merely dependent on PCA-aligned geometries.
> > > >
> > > > To further clarify this point, we are conducting an additional experiment where PCA is removed and replaced with random rotations. This aims to demonstrate that 3D RoPE effectively learns arbitrary directions rather than "memorizing geometries in the PCA frames." To expedite the process, we are utilizing 16 GPUs for training and expect to obtain results within approximately 30 hours.

---

> ### Author Response · Authors · 2024-11-24
>
> We are uncertain about the comment, *"...3D orientation from the combination of 2D rotations (rotations do not commute)."* We suspect there may be a misunderstanding regarding how RoPE works. While we may be mistaken, we provide a more detailed explanation of RoPE below for clarification.
>
> In RoPE, the 2D rotations are not used to encode rotations in a 2D space. Instead, they serve to encode 1D positions $i$ (and $j$) by converting these positions into 2D rotation matrices with angles corresponding to the positions. During the dot product operation between the Query and Key, the relative position $j-i$ is encoded through the composition of two 2D rotation matrices.
>
> For the 3D scenario, since attention mechanisms typically involve multiple heads, we can divide the heads into three groups, with each group encoding the positional information along one of the axes. In this way, 3D RoPE directly encodes $x_{i} - x_{j}$ effectively.

---

> > ### Comment · Reviewer_kPDK · 2024-11-25
> > **More on 3D RoPE**
> >
> > I thank the authors. I don't think there is any misunderstanding here. I am still insisting on the fact that one cannot reconstruct the 3D geometry out of 3 arbitrary chosen 2D projections. For example, please look into the Euler convention - there 12 different canonical ways of how to obtain a 3D rotation out of 2D in-plane rotations.
> > In other words, your approach seems to work in practice, most probably thanks to the PCA normalization, but there are little theoretical guarantees on the 'correct' positional encoding in 3D.

---

> ### Author Response · Authors · 2024-11-25
>
> We believe the reviewer may still have some misunderstandings regarding the mechanism of 3D RoPE.
>
> 3D RoPE does **not** attempt to reconstruct 3D geometry from 2D projections, **nor** does it rely on combining in-plane rotations as in the Euler convention. Instead, 3D RoPE *directly operates on 3D relative positional differences*. For instance, for two 3D points with coordinates $(x_i, y_i, z_i)$ and $(x_j, y_j, z_j)$, 3D RoPE directly encodes the positional deltas $(x_j - x_i, y_j - y_i, z_j - z_i)$ along the three axes. This is achieved **without** involving any 2D projections or 2D angles.
>
> We are uncertain how the reviewer interprets RoPE, so allow us to provide a clearer explanation below.
>
> Let us start with the 1D case. In natural language processing, given two tokens located at positions $x_i$ and $x_j$, the original RoPE mechanism is designed to capture their relative position $x_j - x_i$. This concept is straightforward and widely accepted.
>
> Next, we extend this concept to 2D, as described in [1]. Given two points in 2D space with positions $(x_i, y_i)$ and $(x_j, y_j)$, the goal is to encode their positional differences $x_j - x_i$ and $y_j - y_i$. 2D RoPE achieves this by encoding each positional difference independently. Specifically, in the context of multi-head attention, half of the attention heads are assigned to encode $x_j - x_i$, and the other half to encode $y_j - y_i$.
>
> Similarly, this concept extends naturally to 3D: we encode the relative positional differences along all three axes $(x_j - x_i, y_j - y_i, z_j - z_i)$ by dividing the attention heads into three sets. Each set is dedicated to encoding the positional difference along one axis. This ensures that 3D RoPE directly encodes relative positions in 3D space, without involving any 2D projections or rotations.
>
> From the above explanation, it is clear that the key to 3D RoPE is using three independent sets of 1D RoPE to encode relative positions along the three axes. At no point does it rely on projecting 3D information into 2D space.
>
> We suspect that the reviewer may have misunderstood the role of the 2D rotation matrix used in 1D RoPE. In 1D RoPE, the 2D rotation matrix is used to represent a rotation with an angle corresponding to the token position (e.g., $x_i$ and $x_j$). During the Query-Key dot product operation, the relative position $x_j - x_i$ is encoded through the composition of the two 2D rotation matrices. Thus, in all cases—whether 1D, 2D, or 3D—the 2D rotation matrices are solely used for encoding the positional difference along a single axis and are not related to projecting 3D information into 2D space. For further details, please refer to Equation (2) in the paper or the original RoPE paper [2].
>
> Reference:
>
> [1] Rotary Position Embedding for Vision Transformer.
>
> [2] Enhanced Transformer with Rotary Position Embedding.

---

> > ### Author Response · Authors · 2024-11-26
> >
> > Our additional experiment on “replacing PCA with random rotation” has been completed, and the results are summarized in the following table. The results clearly demonstrate that even without PCA and under randomly rotated 3D inputs, our proposed model achieves strong performance. This confirms that the performance improvement is not attributed to PCA. Moreover, it highlights that our model effectively learns arbitrary 3D directions rather than “memorizing geometries in the PCA frames.”
> >
> > By combining these explicit results with our earlier clarification addressing the reviewer’s misunderstanding regarding 3D RoPE, we hope the reviewer’s concerns have now been fully resolved.
> >
> >
> > | PCA | RoPE | RFF | Random Rotation | R2 ↓ | ZPVE | ↓ Cv ↓ | HOMO ↓ |
> > | --- | ---- | --  | -| ----| ---- | ----| ----|
> > | yes | yes |  yes | no | 2.8363 | 0.0003 | 0.0675 | 0.0017 |
> > | no | yes |  yes | no | 3.3088 | 0.0004 | 0.0708 | 0.0018 |
> > | yes | no |  no  |  no| 3.7104 | 0.0004 | 0.1407 | 0.0022 |
> > | no | yes |  yes  |  yes  | 2.9840 | 0.0003 | 0.0674 |  0.0015  |

---

> > > ### Comment · Reviewer_kPDK · 2024-11-26
> > > **Further random rotations question**
> > >
> > > I thank the authors for the additional experiments. Could you please explain the increased performance compared to the same experiment without random rotations (comparison of the second and forth rows)??
> > >
> > > Could you please also provide numbers at the same precision (the same number of significant digits) for ZPVE and HOMO columns?

---

> > > > ### Author Response · Authors · 2024-11-26
> > > >
> > > > Below is the table providing more precise numbers for ZPVE and HOMO.
> > > > The slight improvement in the results is expected. Specifically, with random rotations, the model is exposed to a diverse range of coordinate systems under varying orientations, which enhances its ability to learn more comprehensively.
> > > > Without random rotations, the model might encounter certain coordinate systems less frequently during training. Consequently, when faced with less common coordinate systems during inference, the model’s performance may degrade. Introducing random rotations ensures a more uniform distribution of coordinate systems,  improving robustness.
> > > >
> > > > | PCA | RoPE | RFF | Random Rotation | R2 ↓ | ZPVE | ↓ Cv ↓ | HOMO ↓ |
> > > > | --- | ---- | --  | -| ----| ---- | ----| ----|
> > > > | yes | yes |  yes | no | 2.8363 | 0.00028366 | 0.0675 | 0.001687503 |
> > > > | no | yes |  yes | no | 3.3088 | 0.00040852 | 0.0708 | 0.00175726 |
> > > > | yes | no |  no  |  no| 3.7104 | 0.000449727 | 0.1407 | 0.002166273 |
> > > > | no | yes |  yes  |  yes  | 2.9840 | 0.00032006 | 0.0674 |  0.00147432  |

---

> > > > > ### Comment · Reviewer_kPDK · 2024-11-26
> > > > > **More on random rotations**
> > > > >
> > > > > I thank the authors for the updated table and the explanation. Could you please explain why initial rotations are not arbitrary chosen? Molecular data does not have canonical orientations, unless you normalize it with PCA, so why additional rotations would magically boost the performance?

---

> > > > > > ### Author Response · Authors · 2024-11-26
> > > > > >
> > > > > > The key difference lies in the training process: our implemented random rotation method applies a different random rotation at each epoch. As a result, with more epochs, the model is exposed to a wider variety of random orientations, enhancing its ability to generalize across different coordinate systems.

---

> > > > > > > ### Comment · Reviewer_kPDK · 2024-11-26
> > > > > > > **On rotational augmentation**
> > > > > > >
> > > > > > > OK, thank you. This technique is also known as rotational augmentation. Basically, the model trained on augmented data performs much better, which indicates the limited expressivity of the relative positional information in the architecture. Please add all these details to the revised version of the manuscript.

---

> > ### Comment · Reviewer_kPDK · 2024-11-26
> > **On 2D and 3D ROPE**
> >
> > I thank the authors for a more detailed explanation, which must be a part of the original manuscript. Following the [1] reference, Rotary Position Embedding for Vision Transformer, I believe your implementation copies the "Axial frequency" approach. The authors of [1] state "The axial frequency is a simple but effective way to expand RoPE for the vision domain. However, it is unable to handle diagonal directions since the frequencies only depend on a single axis." And this issue will be even more problematic in 3D.
> > Also, according to the authors of [1], there is no significant difference between the compared positional encoding approaches in 2D. RPB and APE perform on par with axial RoPE.
> >
> > To remove possible misunderstanding I advise the authors to spend time and experiments on the 3D RoPE part, demonstrating any theoretical guarantees, following the lines of [1] paper, rather than arguing on the "increased performance" without proper validation.

---

> > > ### Author Response · Authors · 2024-11-26
> > >
> > > We are glad that the reviewer now has a correct understanding of 3D RoPE. We will revise the manuscript to make this part clearer and avoid any possible misunderstandings. Additionally, we will include more experiments, including the additional results mentioned above, to further demonstrate the performance of 3D RoPE.
> > >
> > > However, we would like to emphasize that **our main contribution lies in designing a systematic framework that demonstrates  empty space can further enhance molecular representation learning models, rather than proposing a perfect 3D positional encoding**. While 3D RoPE is part of this framework, it is not our primary contribution. We adopted 3D RoPE primarily for its linear complexity. Currently, we use the simplest form of 3D RoPE, which may have minor limitations such as the issue of “diagonal directions.” However, the “mixed frequency” solution proposed in [1] or other more advanced methods can be seamlessly integrated into our framework without affecting our core contribution.
> > >
> > > Regarding APE and RPE: As shown in the above table, our method significantly outperforms APE (the one with "no ROPE + no RFF"). While RPE may achieve better performance, its $O(n^2)$ memory complexity makes it inefficient in our case.
> > >
> > > Overall, we sincerely thank the reviewer for their detailed comments. We hope the reviewer recognizes our efforts and the additional experiments in this discussion period, and we kindly request a reevaluation of our work.

---

### Author Response · Authors · 2024-11-19
**Paper revision and additional experiment**

We sincerely thank all reviewers for their insightful comments and have addressed your questions accordingly. We have also revised our paper based on your feedback. Specifically, we added related work on virtual points used in point clouds and provided clearer explanations in Sec. 3.2 and Sec. 3.3. Additionally, we removed the visualization of the learned representation in response to Reviewer jmPG’s critique, as we acknowledge that this visualization may not offer meaningful insights.

Furthermore, both Reviewer jmPG and Reviewer ApRY noted that our results do not achieve SOTA performance on QM9 compared to baselines like SchNet and PaiNN. We have addressed this concern in our responses. To clarify, the QM9 dataset used in our experiments is a subsampled version containing only **40k** data samples, rather than the full **133k** dataset used in previous works.

To ensure a fair comparison, we conducted an additional experiment. Using the SchNetPack code, we trained SchNet and PaiNN on our downstream dataset and obtained the following results:


| Model          | QM9_mu ↓    | QM9_HOMO  ↓  | QM9_LUMO ↓   | QM9_GAP ↓    | HLM ↓  | MME ↓ | Solu ↓ |
|-----------------|----------|------------|------------|------------|------------|------------|------------|
| **SchNet**     | 0.1554 ± 1.2e-3 | 0.0032 ± 4.3e-5 | 0.0028 ± 6.2e-5 | 0.0045 ± 8.8e-5 | 0.3863 ± 2.2e-2 | 0.3831 ± 2.2e-2 | 0.4419 ± 1e-2 |
| **PaiNN**      | _0.0752_ ± 1.8e-3 | _0.0028_ ± 1.2e-5 | _0.0023_ ± 7.5e-5 | _0.0040_ ± 9.2e-5 | _0.3762_ ± 6.8e-3 | _0.3539_ ± 1.3e-2 | _0.4095_ ± 1.9e-2 |
| **SpaceFormer**| **0.0493** ± 1.3e-3  | **0.0017** ± 1.3e-5 | **0.0019** ± 3.3e-5 | **0.0031** ± 3.1e-5 | **0.2807** ± 1.5e-3 | **0.2794** ± 3.2e-3 | **0.2972** ± 6.9e-3 |

The results clearly demonstrate that SpaceFormer outperforms SchNet and PaiNN across these property prediction tasks.

Lastly, we are actively implementing the energy/force prediction task suggested by Reviewer jmPG and will update the results as soon as they are available.


-----


**Updated**:


We have conducted the additional experiment (energy and force prediction) requested by the reviewer, and the results are presented in the table below. These results clearly demonstrate that our model outperforms both SchNet and PaiNN.

Experimental Setup:

- Dataset: We used QM7-X for this evaluation. To assess the few-shot learning capabilities of the models, we randomly sampled training subsets containing 1k, 5k, 10k, and 20k samples, resulting in four separate experiments. All experiments share the same validation and test datasets, each consisting of 5k randomly sampled samples.
- Baseline Models: We compared our model with SchNet and PaiNN, implemented using SchNetPack (https://github.com/atomistic-machine-learning/schnetpack). All models were **trained using energy loss only**. The force errors were calculated as the gradients of the energy with respect to atomic positions.
- Hyperparameters: All experiments used the same hyperparameter search space (described in Table 8). Each hyperparameter configuration was trained three times with different random seeds, and we report the mean and standard deviation of the results. For all models, the checkpoint with the best validation loss was selected, and the corresponding test set results are reported.


| Model          | # Training Samples | Energy MAE (eV) ↓      | Force MAE (eV/Å) ↓      |
|-----------------|--------------------|-------------------------|-------------------------|
| **SchNet**     | 1k                | 185.1806 ± 6.6565       | 146.4527 ± 5.8946       |
| **PaiNN**      | 1k                | 167.6489 ± 30.9216      | 153.2552 ± 19.6824      |
| **SpaceFormer**| 1k                | **36.9473** ± 7.0779        | **28.6776** ± 2.9315        |
| **SchNet**     | 5k                | 74.5823 ± 4.9014        | 88.3705 ± 3.5321        |
| **PaiNN**      | 5k                | 24.8025 ± 1.9379        | 29.7473 ± 2.4853        |
| **SpaceFormer**| 5k                | **8.3037** ± 1.6529         | **9.5590** ± 2.5805         |
| **SchNet**     | 10k               | 47.7015 ± 1.1859        | 65.1444 ± 2.4222        |
| **PaiNN**      | 10k               | 15.6893 ± 0.7267        | 21.7204 ± 1.2478        |
| **SpaceFormer**| 10k               | **4.3718** ± 0.6450         | **6.8527** ± 1.5114         |
| **SchNet**     | 20k               | 31.2351 ± 1.0555        | 44.2606 ± 0.5122        |
| **PaiNN**      | 20k               | 8.8318 ± 1.4907         | 12.3451 ± 1.7097        |
| **SpaceFormer**| 20k               | **2.0994** ± 0.1833         | **3.3146** ± 0.3387         |

---

> ### Comment · Reviewer_ApRY · 2024-11-23
>
> Thank you for adding the additional results.
> The comment "The results clearly demonstrate that SpaceFormer outperforms SchNet and PaiNN" is based on what exactly? I unfortunately cant see any statistical tests that would underlay that claim - also no uncertainty estimations on the results.
> Could you please provide them as well - otherwise its hard to compare single numbers with each other, not talking any variability into consideration.
>
> Thank you

---

> > ### Author Response · Authors · 2024-11-23
> >
> > Thank you for pointing that out. We realized the standard deviations were missing, and we have now updated the table to include them.

---

### Note · Authors · 2024-12-02

**Comment:**

The discussion period is coming to an end, and we would like to provide a summary of this review discussion. When we first received the comments, we were genuinely surprised by the low score, as we firmly believe that our work makes a meaningful contribution supported by solid empirical experiments. Our decision to engage in this rebuttal was not driven by the expectation of acceptance but rather by a commitment to clearly explain and defend our work.

Objectively, we believe we have addressed most of the concerns raised by the reviewers. Specifically, we have responded to a total of 29 questions or weaknesses raised by 4 reviewers. While the majority of our responses addressed their concerns, there are some additional comments:

- **Misunderstanding of a component**: One reviewer initially evaluated our work based on a misunderstanding. During the discussion, we clarified this point and revised the paper accordingly. We believe that misunderstanding has been resolved.
- **Comparison with neural potential models (NPMs)**: One reviewer argued that we should compare our model to NPMs, which we believe is beyond the scope of this paper. Nevertheless, we conducted additional experiments to address this concern. Unfortunately, the reviewer questioned the validity of these results and remained unconvinced. Additionally, they raised concerns about the "iso-FLOP" comparison with baselines but did not provide further justification.
- **Data split concerns**: One reviewer argued that our downstream tasks were not based on scaffold splits, questioning the validity of our results. In response, we demonstrated that our OOD split is very similar to scaffold splits for 5 out of 6 datasets, with only 0.4–1.6% scaffold overlap. For the remaining dataset (QM9), we conducted an additional experiment using the scaffold split, which confirmed that the performance ranking is consistent with our original split method.

We greatly appreciate the reviewers' time and effort in evaluating our work and contributing to this discussion. We recognize that “addressing concerns” does not necessarily translate to a “positive score,” but we believe we have achieved our primary goal: to clearly explain and defend our work.

As such, we have decided to withdraw our submission. We sincerely thank all the reviewers again for their feedback and engagement throughout the discussion period.

**Withdrawal Confirmation:**

I have read and agree with the venue's withdrawal policy on behalf of myself and my co-authors.